# Effects of Household Processing on Residues of the Chiral Fungicide Mandipropamid in Four Common Vegetables

**DOI:** 10.3390/ijerph192315543

**Published:** 2022-11-23

**Authors:** Shiyin Mu, Li Dou, Yu Ye, Du Chi, Kankan Zhang

**Affiliations:** State Key Laboratory Breeding Base of Green Pesticide and Agricultural Bioengineering, Key Laboratory of Green Pesticide and Agricultural Bioengineering, Ministry of Education, Guizhou University, Guiyang 550025, China

**Keywords:** mandipropamid, vegetable, removal efficiency, processing factor, health risk

## Abstract

The study aimed to detect the content of mandipropamid enantiomers in unprocessed and processed tomato, cucumber, Chinese cabbage, and cowpea samples and assess the health risks to Chinese consumers. Data showed that washing and soaking with an acidic solution reduced the mandipropamid residue from vegetable samples by 54.1–82.2%. The pickling process resulted in a 6.2–65.2% loss of mandipropamid from cucumber, Chinese cabbage, and cowpea samples. Peeling and juicing were the best removing techniques for mandipropamid residues in tomato and cucumber (removal rate (RR) value > 91%), and cooking for 5 min could effectively reduce the levels of mandipropamid in Chinese cabbage and cowpea (RR values of 81.4–99.7%). The values of processing factor for the processed vegetable samples are all less than one. No significant enantioselectivity of mandipropamid was found in the vegetables during processing. Health risk data showed that samples of four types of mandipropamid-contaminated vegetables were safe for consumption after processing.

## 1. Introduction

Food safety is a growing area of global concern due to its direct impact on human health and the environmental ecosystems. While agricultural crops provide the nutrients humans need, they are vulnerable to the threats caused from fungi, bacteria, viruses, insects, and weeds. Pesticides are promoted and applied in agricultural culture and production to control threats and improve the safety of agricultural crops. However, pesticide residues have been found in almost all agricultural crops (both raw and processed), which might cause human adverse reactions and diseases, such as nausea, headaches, and cardiovascular disease [1]. The safety of vegetables, a major component of agricultural crops, is also a major concern for Chinese consumers. In China, vegetables provide various vitamins, minerals, and dietary fibres and are indispensable constituents of the daily family diet [2]. Currently, pesticides are widely used during the growing and production stages of vegetables to ensure good quality and sufficient yield. Some pesticides are absorbed by vegetable tissues and transported through the metabolic system within the vegetable. Some pesticides only attach to the surface of vegetables and protect the vegetable from the fungi and insects of direct contact [3]; however, the use of pesticides has contributed to the potential contamination threat to vegetables and their growing environs.

The most common pathway of pesticide ingestion in humans is through food ingestion, where most foods are consumed after a range of culinary and processing treatments. Several studies have demonstrated that food processing techniques may reduce or increase pesticide concentrations in both raw and processed agricultural crops. Household processing techniques, including washing [4], pickling [5], peeling [6], boiling [7], drying [8], and juicing [9], are the most common approaches for changing (eliminating or concentrating) the residues of pesticides in agricultural products. In accordance with the food processing data presented in the FAO/WHO Joint Meeting on Pesticide Residues (JMPRs) report, differences in pesticide residues occur in crops and their products [10]. In addition, some researchers illustrated that the effect of the processing method on reducing pesticide residues might be influenced by the physicochemical properties of pesticides, such as solubility, volatility, octanol/water distribution coefficient (*K*_ow_), and mode of action [11]. For instance, washing and peeling can effectively remove the residues of chlorpyriphos and ethylparathion in tomato, probably due to the log *K*_ow_ values of 4.7 and 3.8, respectively, which inhibited their transport in plant tissue and located on the outer epidermis of crops [9]. When boiling and steaming were used in processing rice products, volatilization and thermal degradation caused the reduction in pesticide residues in the range of 20.7–100% [12].

Mandipropamid is a chiral systemic fungicide with a broad bactericidal spectrum and can effectively prevent most foliar oomycete pathogens. The market sales of mandipropamid have been growing in the last half decade [13]. As applications increase, potential contamination from mandipropamid residues in the ecosystem became more apparent. A few studies have reported the dissipation and residue distribution of mandipropamid in vegetables and environmental matrices at the enantiomer level [11,14,15], whereas a lack of household processing techniques and data exists. On the basis of our previous research [16], this work evaluated the efficiency of several household processing approaches on removing the residues of mandipropamid enantiomers from four kinds of vegetable samples (tomato, cucumber, Chinese cabbage, and cowpea), calculated and analysed the processing factor (PF) values, and assessed the health risk of mandipropamid for different groups of Chinese consumers.

## 2. Materials and Methods

### 2.1. Chemicals and Reagents

The analytical standard of racemic mandipropamid (purity, 98.6%) was purchased from J&K Scientific Ltd. (Beijing, China), and the formulation of suspension concentrate (SC) (23.4%) was obtained from Syngenta Nantong Crop Protection Co., Ltd. (Nantong, China). Chromatographic grade acetonitrile (ACN), methanol (MeOH), ammonium formate (AF) and formic acid (FA) were bought from Thermo Fisher Scientific (Waltham, USA). Analytical grade anhydrous sodium acetate (NaAc), magnesium sulfate (MgSO_4_), sodium chloride (NaCl), sodium carbonate (Na_2_CO_3_), and acetic acid (AA) were purchased by Tianjin Zhiyuan Reagent Co., Ltd. (Tianjin, China). Primary secondary amine (PSA) and octadecylsilane (C_18_) were obtained from the Bonna-Agela Company (Tianjin, China). Distilled water and 0.22 µm nylon syringe filters were bought from Watson Group Ltd. (Dongguan, China) and PeakSharp Technologies (Beijing) Co., Ltd. (Beijing, China), respectively.

### 2.2. Instrumentation and Sample Pretreatment

Liquid chromatography with tandem mass spectrometry (LC–MS/MS) was applied for the determination of mandipropamid enantiomers. Chromatographic separation was performed on a Daicel Chiralcel OZ-3R column (150 × 2.1 mm i.d., particle size 3 µm, Daicel Chiral Technologies Co., Ltd., Shanghai, China) installed on a Shimadzu LC-20AD XR LC system (Shimadzu Corporation, Kyoto, Japan), and quantitative analysis was conducted on an AB Sciex API 4000 Q-Trap quadrupole MS system (Applied Biosystems, Foster City, CA, USA). The extraction solution for vegetable samples was ACN with 1% AA, the extraction method was vortex extraction, and the purification sorbent was a different combination of PSA, C_18_, and MgSO_4_, while the brine samples were filtered through gravity filter paper and a 0.22 μm nylon syringe filter and then directly injected into the instrument. The detailed liquid chromatographic conditions, mass spectrometric parameters, and pre-treatment procedure were the same as those in our previous study [16].

### 2.3. Household Processing Techniques

In the household processing experiment, all four kinds of vegetable samples were gathered on the 1st day after the last application in the field trials, which were carried out in 2021 with the application of 23.4% mandipropamid SC at a spray dosage of 140.4 g a.i./hm^2^ (the recommended high dosage) three times with an interval of 7 d. After different processing procedures, the vegetable samples were chopped into small pieces with an average size of approximately 1 cm^3^ and then homogenized before pre-treatment. Figure 1 demonstrates the scheme of household processing techniques for four types of vegetable samples. Three replicates were performed in each treatment. The detailed steps of the different processes are listed in the following subsections.

#### 2.3.1. Washing and Soaking

Four kinds of non-toxic solutions, including 10% (*v*/*v*) acidic solution (AA solution), 10% (*w*/*v*) solution (NaCl solution), 10% alkaline (*w*/*v*) solution (Na_2_CO_3_ solution), and tap water, were chosen to assess the effect of washing on the residues mandipropamid enantiomers in tomato, cucumber, Chinese cabbage, and cowpea. Mandipropamid-contaminated vegetable samples collected in the field trials in August 2021, which were directly transferred to the lab and prepared as lab samples, were washed in different solutions for 1 min [17,18]. In this study, four types of contaminated vegetable samples were soaked for 10 min in the above four solutions [19]. The processed samples were extracted and detected using the analysis method described in Section 2.2.

#### 2.3.2. Pickling

The contaminated cucumber samples were cut into small pieces with an average size of approximately 1 cm^3^, and the contaminated Chinese cabbage and cowpea samples were drained as the raw material of sauerkraut [20,21,22,23]. The pretreated vegetable samples were pickled in 10% NaCl solution in a 5 L fermentation tank stored in the dark at 25 °C. The pickled vegetable and brine samples were collected at 7 time points (0, 1, 3, 5, 7, 14, and 21 d), and all the samples were stored at −20 °C for further analysis.

#### 2.3.3. Peeling and Juicing

The mandipropamid-contaminated cucumber or tomato samples were peeled with a knife of 1 mm approximate thickness at room temperature [24,25]. The peel sample and a partial pulp sample were subsequently detected. Another part of the pulp sample was homogenized to obtain the juice and puree samples.

#### 2.3.4. Cooking

Part of the mandipropamid-contaminated Chinese cabbage or cowpea samples were boiled in 93 °C water for 1, 3, and 5 min, and another part was steamed over boiling water (96 °C) for 1, 3, and 5 min [26,27]. The samples were then immediately analyzed as a whole for remaining mandipropamid enantiomers.

#### 2.3.5. Data Calculation

The removal rate (RR, %) is one parameter used to assess the effect of processing techniques on the change in pesticide residues in processed products [28]. The values are calculated by the equation RR = (*C*_1_− − *C*_2_)/*C*_1_ × 100%, where *C*_1_ and *C*_2_ is the residue concentration (µg/kg) in unprocessed and processed products, respectively. RR values of <100% indicate a decrease in pesticide residues, while RR values of >100% indicate an increase in pesticide residues in processed products. PF value, the ratio of *C*_2_ and *C*_1_, is also used to measure the effectiveness of a processing approach [2]. As the PF values < 1, the process technique (washing or boiling) can reduce pesticide residues in vegetables [25]. A PF value of 1 indicates that the method has no effect on pesticide residues. However, when PF values are more than 1, the concentrations of pesticides in vegetables increase after certain treatments, such as juicing [29]. Enantiomeric fraction (EF) values are calculated by the equation EF = *C_R_*/(*C_R_* + *C_S_*), where *C_R_* and *C_S_* are the residue levels of *R*- and *S*-enantiomer, respectively, and applied to evaluate the potential enantioselectivity of mandipropamid in vegetable samples after different processing treatments. As 0 < EF < 0.5, a preferential removal efficiency was conducted for *R*-enantiomer. An EF value of 0.5 indicates no enantioselectivity. Moreover, the *S*-enantiomer is preferentially removed for EF values in the range of 0.5–1 [21,30].

### 2.4. Health Risk Estimation

Hazard quotient (HQ) values, including long-term (HQ_Chronic_) and short-term (HQ_Acute_), are used to assess the health exposure risk of mandipropamid in vegetable samples after different processing procedures [29]. The JMPR evaluation report demonstrates that the acute reference dose (ARfD) value of mandipropamid is unnecessary and that the acute risk can be negligible to humans. Therefore, the long-term risk should be assessed for mandipropamid and the HQ_Chronic_ is calculated as the ratio of international estimated daily intake (IEDI) and acceptable daily intake (ADI). The IEDI value is obtained from the equation IEDI = (FI × RL)/b.w., where FI is the food intake data, RL is the residue level of mandipropamid in processed food samples, and b.w. is the body weight. The ADI value for mandipropamid is 0.2 mg/kg b.w. [14].

### 2.5. Statistical Analysis

All the processing experiments were performed three times. Excel 2010 (Microsoft Corporation, Washington, DC, USA) was applied to the data analysis. Through Duncan’s multiple range test, the data were statistically evaluated by one-way analysis of variance (ANOVA, *p* = 0.05) with SPSS ver. 24.0 statistical software (SPSS Inc., Chicago, IL, USA).

## 3. Results

### 3.1. Method Validation

The analytical method of mandipropamid enantiomers in vegetable samples was quantified by the approach of calibration curves and validated by limit of detection (LOD), limit of quantitation (LOQ), and recovery and relative standard deviation (RSD). Based on the reported determination approach [16], the LODs and LOQs of two mandipropamid enantiomers in the whole part, skin, pulp, puree, and juice of different vegetable samples were 0.7 and 2.5 μg/kg, respectively. In Appendix A, average recoveries of *R*-enantiomer and *S*-enantiomer in different parts were 82.6–99.6% for tomato, 88.7–100.8% for cucumber, 96.0–101.0% for Chinese cabbage, and 89.8–98.8% for cowpea, with RSDs of 0.2–13.1% in the spiked level range of 2.5–500 μg/kg.

### 3.2. Effect of Household Processing Techniques

#### 3.2.1. Washing and Soaking

The data on the effects of washing and soaking on the residues of mandipropamid enantiomers and racemate in four types of vegetable samples are shown in Table 1, Table 2, Table 3 and Table 4. For example, after washing in tap water, the *R*-enantiomer, *S*-enantiomer, and *rac*-mandipropamid in cucumber samples had RR values of 23.6, 23.9, and 23.8%, respectively. The RR values were approximately 47% higher in the cucumber samples washed in NaCl solution and Na_2_CO_3_ solution, and the corresponding RR values were more than 2.2 times higher in the cucumber samples washed in AA solution. Additionally, the PF values of washing in AA solution were significantly lower than those in the other three solutions (*p* < 0.001, Table 1 and Table 2). For cowpea, the RR values of the three analytes in samples washed in AA solution were significantly higher by approximately three times than those in samples washed in tap water (*p* < 0.001, Table 3 and Table 4). In Table 1 and Table 2, the RR values of the three analytes in tomato and cucumber samples soaked in 10% AA solution (75.3–82.2%), 10% Na_2_CO_3_ solution (71.5–77.0%), 10% NaCl solution (69.9–73.1%), and tap water (57.8–60.1%) were significantly higher than those in samples washed in the four solutions (*p* < 0.001). The data in Table 3 and Table 4 showed that after soaking, the RR values of *R*-enantiomer (56.2–76.9%), *S*-enantiomer (57.2–76.8%), and *rac*-mandipropamid (56.7–76.8%) were significantly higher than those after washing, and the PF values for soaking (0.231–0.438) were significantly lower than those for washing (0.444–0.820) (*p* < 0.001).

#### 3.2.2. Pickling

Figure 2 shows the concentration, RR, and PF values of mandipropamid in samples of pickled cucumber, Chinese cabbage, and cowpea at different intervals. Residues of mandipropamid were significantly reduced in the pickled cucumber samples, with a RR value of 17.8–64.5% during the 21-day pickling period, followed by the pickled cowpea samples, with a RR value of 8.1–51.3%. In pickled Chinese cabbage samples, the mandipropamid enantiomers and racemate showed relatively low removal rates. The PF data (Figure 2) are also demonstrated the different removal efficiencies for mandipropamid in different vegetable samples after pickling. For example, the PF values of pickling decreased significantly from 0.796 to 0.356, from 0.883 to 0.486, and from 0.935 to 0.658 for the *R*-enantiomer in cucumber, cowpea, and Chinese cabbage samples, respectively (*p* < 0.001). Moreover, some mandipropamid residues were transferred from vegetables into brine (pickling solution). In Appendix A, the concentrations of mandipropamid enantiomers and racemate in brine samples increased from the first day to the seventh day and then decreased in the following 14 days. 

#### 3.2.3. Peeling and Juicing

As shown in Table 5, the percentages of mandipropamid residues were <10% in tomato pulp samples and approximately 1% in cucumber pulp samples after peeling. More than 90% of mandipropamid is found in vegetable skins. After peeling, the tomato and cucumber pulp samples are cut into quarters and separated into puree and juice. The RR values of mandipropamid residues in puree and juice samples are 97.1–97.9% for tomato and 99.4–99.5% for cucumber. The corresponding PF values are in the range of 0.005–0.029.

#### 3.2.4. Cooking

The effect of the cooking process on the mandipropamid residues in Chinese cabbage and cowpea samples was investigated for two treatments: boiling and steaming. After boiling for 1, 3, and 5 min, the RR values of mandipropamid significantly increased from 44.6 to 99.0% in Chinese cabbage, with PF values of 0.009–0.554, and from 65.1 to 99.7% in cowpea, with PF values of 0.003–0.349 (*p* < 0.001, Figure 3). After steaming, the concentrations of *R*-enantiomer significantly decreased from 5969 to 1108 µg/kg in Chinese cabbage, with decreasing PF values from 0.793 to 0.186 after steaming for 5 min (*p* < 0.001, Figure 3). For cowpea, the difference in removal efficiency between boiling and steaming was not significant. The results show that these two cooking treatments also effectively removes mandipropamid residues from Chinese cabbage and cowpea.

#### 3.2.5. Potential Enantioselectivity

After washing and soaking in different solutions, the difference between the *R*- and *S*-enantiomer residues in the four vegetables remained nearly constant, with EF values close to 0.5 (Figure 4). No significant enantioselectivity was also found in the pickled vegetable samples. Although after peeling or cooking, a few EF values were changed considerably, such as EF values increased to 0.565 in tomato skin samples and to 0.541 in cucumber skin samples or EF values decreased to 0.447 after steaming for 5 min and to 0.382 after boiling for 5 min in cowpea samples (Figure 4), the reduction in mandipropamid enantiomers was not different in most cases.

### 3.3. Health Risk

The HQ values were calculated and used to assess the health risks of mandipropamid in processed vegetable samples for different groups of Chinese consumers. For the chronic risk assessment, HQ_chronic_ values were calculated and are shown in Table 6, Table 7, Table 8 and Table 9. After different processing treatments, the HQ_chronic_ values of *rac*-mandipropamid were <2% for the tomato samples, <8% for the cucumber samples, <40% for the Chinese cabbage samples, and <11% for the cowpea samples.

## 4. Discussion

Based on the method validation data (Appendix A), the developed extraction and detection method can be employed in the following processing experiments. Due to log *K*_ow_ of 3.829, most mandipropamid residues remained in the surface of vegetables [9] and washing can effectively remove those, which is consistent with our obtained results that washing in different aqueous solutions (including acidic, basic, neutral solutions, and tap water) can remove most mandipropamid residues from tomato, cucumber, Chinese cabbage, and cowpea samples. Among the four solutions, tap water had the lowest reduction efficiency, NaCl solution and Na_2_CO_3_ solution were the next lowest, and AA solution was the most effective in reducing mandipropamid residues from all four types of vegetable samples. Similar results have been reported for other pesticides in a number of previous studies. For example, Kin and Huat found that for cucumber and strawberry samples, the removal efficiencies of washing in 10% AA solution (RR values of 44–70%) were more excellent than those in 10% Na_2_CO_3_ solution (RR values of 30–50%), 10% NaCl solution (RR values of 23–40%), and tap water (RR values of 10–20%) for eight pesticides [18]. Soliman reported that washing in AA solution had the best effect in reducing pesticide residues from potato samples [31]. Compared with washing (PF values of 0.413–0.764), soaking increased the reduction amount of mandipropamid enantiomers in tomato and cucumber samples with a lower PF value of 0.178–0.422, which illustrated that soaking had a better removing efficiency. Similarly, soaking was also found to be more effective than washing in removing mandipropamid residues from Chinese cabbage and cowpea samples. In the investigation of the reduction effect of soaking for pesticide residues in potato samples, Zohair found higher RR values (>75%) compared with the corresponding data of washing [19]. The possible reason is that soaking provides a relatively long-term contact of the aqueous solution with the vegetable sample. It was shown that washing and soaking effectively removed the mandipropamid residue from the four vegetables.

In the pickling trial, the concentrations of mandipropamid were slightly reduced in cucumber, Chinese cabbage, and cowpea samples over the pickling time. Microbiological degradation may be the main pathway for removing pesticide residues from vegetables in the pickling process [21]. Different reduction efficiencies were obtained for the analyte in different vegetables, which might be related to the characteristics of the vegetables [20,22], such as specific bacteria generated after pickling different vegetables [5]. Based on the chemical properties of pesticides and environmental conditions, peeling and juicing are two efficient approaches for removing pesticide residues from vegetable products [32]. The results showed that the peeling technique had a significant effect in removing mandipropamid residues from tomato and cucumber samples, with the skins accumulating significant amounts of mandipropamid. The present data are in agreement with some previous studies, where the RR value of peeling was 63% for chlorpyrifos residues in tomato sample with a PF value of 0.09 [33], and the RR value for dichlorvos residues in cucumber sample was 57% after peeling [34]. The reason was possibly related to the mode of action of the analyte: before the systemic interaction, the contact of mandipropamid and vegetable occurred and almost remained on the surface of the skin [3]. Although no significant difference was observed between the RR and PF values in juice and puree, the lower water solubility of mandipropamid (4.2 mg/L) made it difficult to be transported into tomato or cucumber juice [24]. Boiling was also an effective processing technique for reducing the mandipropamid residue in Chinese cabbage and cowpea. A similar process was found in the study of Kontou et al. [26] and Rowayshed et al. [35], where boiling removed more than 70% of maneb residues from tomato samples and removed >90% of carbamate pesticide residues from eggplant, respectively. Compared with boiling, steaming may be a less effective treatment for reducing pesticide residues in vegetables, as only high temperatures and water vapour can cause pesticide loss. However, after steaming, the residues of mandipropamid in Chinese cabbage and cowpea decreased to a great extent, which possibly contributed to the character of vegetables [3]. Thus, the above results demonstrated that significant reductions in mandipropamid were found in tomato and cucumber samples after peeling and juicing, and in Chinese cabbage and cowpea samples after cooking, and that the processing techniques have a significant removal effect on the mandipropamid residue in vegetables.

Based on the EF values (Figure 4), the distribution of mandipropamid in tomato, cucumber, Chinese cabbage and cowpea was not significantly enantioselective after the above four process treatments, indicating that household processing may not induce enantioselectivity of mandipropamid residues in vegetables. The maximum residue limit (MRL) value is used to evaluate the safety of pesticide residues in vegetables [16]. The MRL values of mandipropamid were 300, 200, and 25,000 µg/kg for tomato, cucumber, and Chinese cabbage in China, respectively [36], and 1000 µg/kg for beans with pods in the European Union [37]. According to the concentration data of *rac*-mandipropamind in the processed vegetable samples, the tomato sample was safe after washing, soaking, peeling, and juicing; the cucumber sample was safe after peeling and juicing; the Chinese cabbage sample was safe after all kinds of processing; and the cowpea sample was safe after cooking (>1 min). In addition, all the HQ_chronic_ values were less than 100%, indicating that the health risks of the four mandipropamid-contaminated vegetable samples are acceptable for Chinese consumers after processing. The residue and health risk assessment results illustrated the impressive removal efficiency of the processing method for mandipropamid residues in tomato, cucumber, Chinese cabbage, and cowpea samples.

## 5. Conclusions

In this study, the effects of different processing techniques, including washing, soaking, pickling, peeling, juicing, boiling, and steaming, on mandipropamid residues were investigated in four types of vegetables. The results showed that the concentration of mandipropamid decreased after the above processing procedure. The addition of 10% AA, 10% Na_2_CO_3_, and 10% NaCl significantly increased the removal efficiency of mandipropamid compared with tap water washing and soaking. The PF values (0.178–0.459) demonstrated that washing and soaking with the acidic solution reduced the mandipropamid residue more than the other three solutions. The decrease in PF from 0.938 to 0.348 illustrated that pickling is a moderate processing technique for reducing mandipropamid residues from vegetable samples. Peeling and juicing were able to effectively remove the mandipropamid residue from the tomato and cucumber samples. For Chinese cabbage and cowpea, the concentrations of mandipropamid were reduced by >81% after boiling and steaming for 5 min. After these post-processing treatments, there was no significant enantioselectivity of mandipropamid in the processed vegetable samples. HQ values < 100% demonstrated that the health risks of mandipropamid in four types of processed vegetables is negligible for Chinese consumers. The results not only provide information about the impact of mandipropamid on food safety, but also help screen contaminated agricultural products for processing methods.

## Figures and Tables

**Figure 1 ijerph-19-15543-f001:**
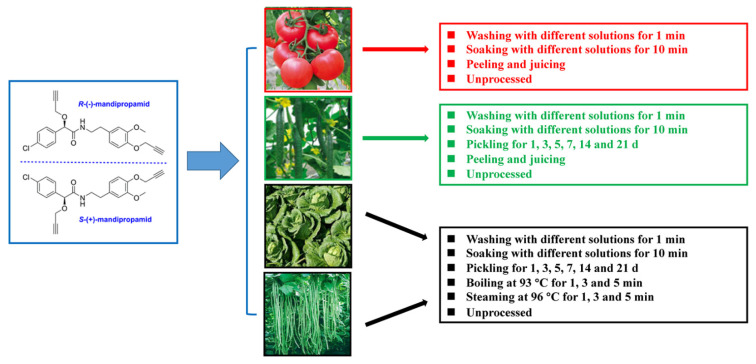
Scheme for assigning processing to four types of mandipropamid-treated vegetable samples.

**Figure 2 ijerph-19-15543-f002:**
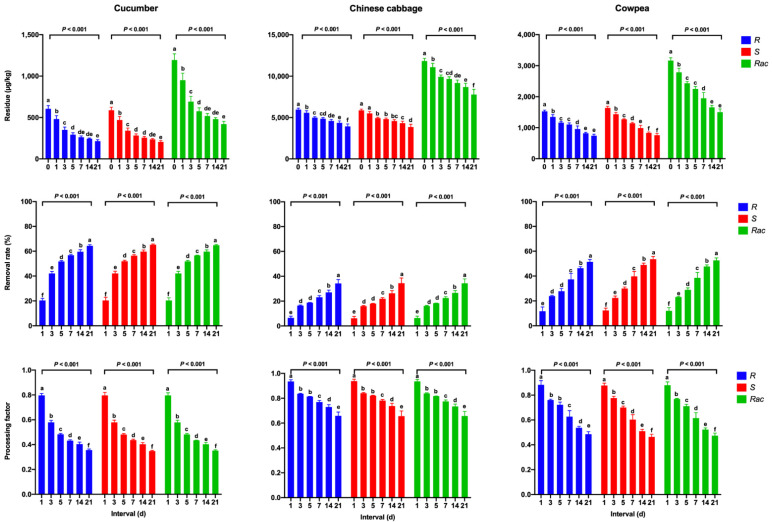
Residues, removal rates (RRs) of mandipropamid enantiomers, and racemate and processing factors (PFs) for different intervals after pickling in cucumber, Chinese cabbage, and cowpea samples. Error bars represent the standard deviation (SD) of triplicates. Each lowercase letter indicates significant differences between the seven sampling intervals (*p* ≤ 0.05, Duncan’s multiple range test).

**Figure 3 ijerph-19-15543-f003:**
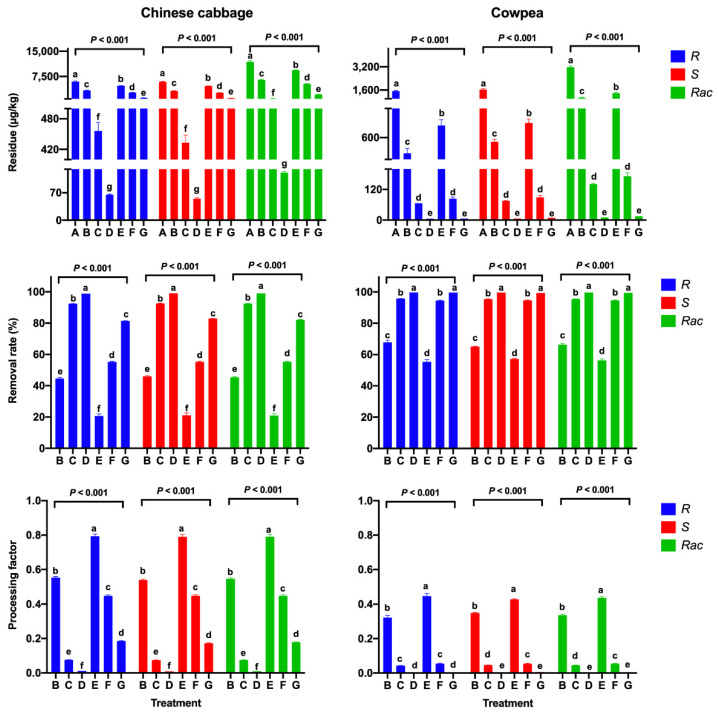
Residues and RRs of mandipropamid enantiomers and racemate and PFs for different cooking steps in Chinese cabbage and cowpea samples (A: unprocessed; B: boiled for 1 min; C: boiled for 3 min; D: boiled for 5 min; E: steam for 1 min; F: steam for 3 min; G: steam for 5 min). Error bars represent the SD of triplicates. Each lowercase letter indicates significant differences between the seven sampling intervals (*p* ≤ 0.05, Duncan’s multiple range test).

**Figure 4 ijerph-19-15543-f004:**
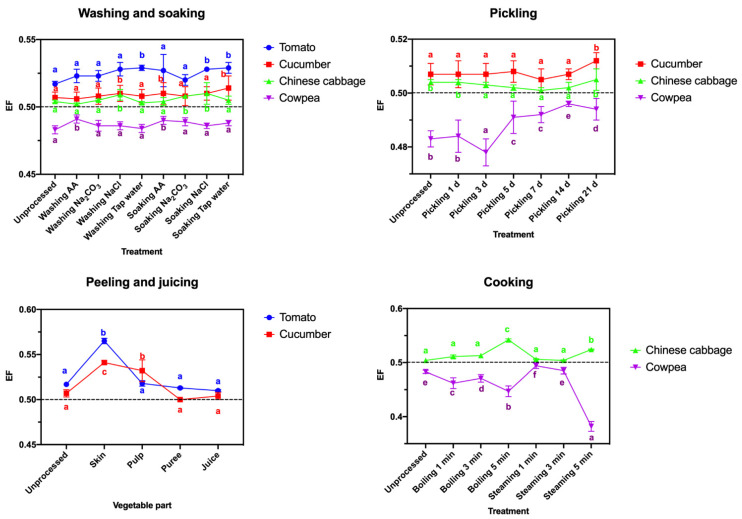
EF values of mandipropamid in four types of vegetable samples after different processing treatments. Each lowercase letter indicates significant differences between the different treatments (*p* ≤ 0.05, Duncan’s multiple range test).

**Table 1 ijerph-19-15543-t001:** Residue level, removal rate (RR) and processing factor (PF) values of mandipropamid in unprocessed, washing, and soaking tomato samples.

Analyte	Treatment	Solution	Average Value ± SD (*n* = 3)					
			Residue Level (µg/kg)	*P* _1_	RR (%)	*P* _2_	PF	*P* _3_
*R*	Unprocessed	/	195 ± 5 a	<0.001	/	<0.001	/	<0.001
	Washing	AA	82 ± 2 d		57.8 ± 1.0 d		0.422 ± 0.010 d	
		Na_2_CO_3_	91 ± 8 d		53.4 ± 3.0 e		0.466 ± 0.030 c	
		NaCl	105± 10 c		46.1 ± 3.8 f		0.539 ± 0.038 b	
		Tap water	123 ± 2 b		37.1 ± 0.4 g		0.629 ± 0.004 a	
	Soaking	AA	36 ± 5 f		81.5 ± 2.4 a		0.185 ± 0.024 g	
		Na_2_CO_3_	45 ± 4 f		76.8 ± 1.5 b		0.232 ± 0.015 f	
		NaCl	55 ± 0 e		72.0 ± 0.6 c		0.280 ± 0.006 e	
		Tap water	82 ± 0 d		57.8 ± 1.2 d		0.422 ± 0.012 d	
*S*	Unprocessed	/	182 ± 3 a	<0.001	/	<0.001	/	<0.001
	Washing	AA	75 ± 2 de		58.7 ± 0.6 d		0.413 ± 0.006 d	
		Na_2_CO_3_	83 ± 7 d		54.5 ± 3.1 e		0.455 ± 0.031 c	
		NaCl	94 ± 9 c		48.3 ± 4.0 f		0.517 ± 0.040 b	
		Tap water	109 ± 3 b		40.0 ± 0.7 g		0.600 ± 0.007 a	
	Soaking	AA	32 ± 6 g		82.2 ± 3.1 a		0.178 ± 0.031 g	
		Na_2_CO_3_	42 ± 4 f		77.0 ± 2.1 b		0.230 ± 0.021 f	
		NaCl	49 ± 0 f		73.1 ± 0.3 c		0.269 ± 0.003 e	
		Tap water	73 ± 1 e		59.6 ± 0.2 d		0.404 ± 0.002 d	
*Rac*	Unprocessed	/	376 ± 8 a	<0.001	/	<0.001	/	<0.001
	Washing	AA	157 ± 4 de		58.2 ± 0.6 d		0.418 ± 0.006 d	
		Na_2_CO_3_	173 ± 15 d		53.9 ± 3.0 e		0.461 ± 0.030 c	
		NaCl	199 ± 18 c		47.1 ± 3.8 f		0.529 ± 0.038 b	
		Tap water	231 ± 5 b		38.5 ± 0.2 g		0.615 ± 0.002 a	
	Soaking	AA	68 ± 11 g		81.8 ± 2.7 a		0.182 ± 0.027 g	
		Na_2_CO_3_	87 ± 9 f		76.9 ± 1.8 b		0.231 ± 0.018 f	
		NaCl	103 ± 1 f		72.5 ± 0.4 c		0.275 ± 0.004 e	
		Tap water	155 ± 0 e		58.7 ± 0.7 d		0.413 ± 0.007 d	

*P*_1_, *P*_2_, and *P*_3_ represent residue level, RR, and PF, respectively. Different lower-case letters indicate statistical significance between different treatments for mandipropamid enantiomers and racemate by Duncan’s multiple range test (*p* ≤ 0.05).

**Table 2 ijerph-19-15543-t002:** Residue level, RR, and PF values of mandipropamid in unprocessed, washing and soaking cucumber samples.

Analyte	Treatment	Solution	Average Value ± SD (*n* = 3)					
			Residue Level (µg/kg)	*P* _1_	RR (%)	*P* _2_	PF	*P* _3_
*R*	Unprocessed	/	605 ± 39a	<0.001	/	<0.001	/	<0.001
	Washing	AA	277 ± 13 de		54.2 ± 0.8 d		0.458 ± 0.008 c	
		Na_2_CO_3_	316 ± 21 cd		47.9 ± 0.9 e		0.521 ± 0.009 b	
		NaCl	325 ± 17 c		46.3 ± 0.7 e		0.537 ± 0.007 b	
		Tap water	463 ± 40 b		23.6 ± 2.6 f		0.764 ± 0.026 a	
	Soaking	AA	150 ± 4 f		75.1 ± 1.0 a		0.249 ± 0.010 f	
		Na_2_CO_3_	172 ± 10 f		71.5 ± 0.3 b		0.285 ± 0.003 e	
		NaCl	182 ± 12 f		69.9 ± 0.5 b		0.301 ± 0.005 e	
		Tap water	248 ± 18 e		59.0 ± 0.8 c		0.410 ± 0.008 d	
*S*	Unprocessed	/	588 ± 36 a	<0.001	/	<0.001	/	<0.001
	Washing	AA	270 ± 9 d		54.1 ± 1.2 d		0.459 ± 0.012 c	
		Na_2_CO_3_	306 ± 13 c		48.0 ± 1.0 e		0.520 ± 0.010 b	
		NaCl	312 ± 13 c		46.9 ± 1.1 e		0.531 ± 0.011 b	
		Tap water	448 ± 32 b		23.9 ± 1.4 f		0.761 ± 0.014 a	
	Soaking	AA	144 ± 8 f		75.4 ± 0.3 a		0.246 ± 0.003 f	
		Na_2_CO_3_	167 ± 8 f		71.7 ± 0.3 b		0.283 ± 0.003 e	
		NaCl	175 ± 9 f		70.3 ± 0.3 b		0.297 ± 0.003 e	
		Tap water	235 ± 21 e		60.1 ± 1.5 c		0.399 ± 0.015 d	
*Rac*	Unprocessed	/	1194 ± 75 a	<0.001	/	<0.001	/	<0.001
	Washing	AA	546 ± 22 d		54.2 ± 1.0 d		0.458 ± 0.010 c	
		Na_2_CO_3_	621 ± 34 c		47.9 ± 0.7 e		0.521 ± 0.007 b	
		NaCl	637 ± 29 c		46.6 ± 0.9 e		0.534 ± 0.009 b	
		Tap water	911 ± 72 b		23.8 ± 2.0 f		0.762 ± 0.020 a	
	Soaking	AA	295 ± 11 e		75.3 ± 0.7 a		0.247 ± 0.007 f	
		Na_2_CO_3_	339 ±17 e		71.6 ± 0.3 b		0.284 ± 0.003 e	
		NaCl	357 ± 22 e		70.1 ± 0.3 b		0.299 ± 0.003 e	
		Tap water	483 ± 39 d		59.5 ± 1.2 c		0.405 ± 0.012 d	

*P*_1_, *P*_2_, and *P*_3_ represent residue level, RR, and PF, respectively. Different lower-case letters indicate statistical significance between different treatments for mandipropamid enantiomers and racemate by Duncan’s multiple range test (*p* ≤ 0.05).

**Table 3 ijerph-19-15543-t003:** Residue level, RR, and PF values of mandipropamid in unprocessed, washing, and soaking Chinese cabbage samples.

Analyte	Treatment	Solution	Average Value ± SD (*n* = 3)					
			Residue Level (µg/kg)	*P* _1_	RR (%)	*P* _2_	PF	*P* _3_
*R*	Unprocessed	/	5969 ± 164 a	<0.001	/	<0.001	/	<0.001
	Washing	AA	2689 ± 93 de		55.0 ± 0.3 d		0.450 ± 0.003 d	
		Na_2_CO_3_	2844 ± 137 d		52.4 ± 1.0 e		0.476 ± 0.010 c	
		NaCl	3145 ± 208 c		47.4 ± 2.2 f		0.526 ± 0.022 b	
		Tap water	3807 ± 128 b		36.2 ± 0.4 g		0.638 ± 0.004 a	
	Soaking	AA	1382 ± 72 g		76.9 ± 0.6 a		0.231 ± 0.006 g	
		Na_2_CO_3_	1478 ± 41 fg		75.2 ± 0.0 a		0.248 ± 0.000 g	
		NaCl	1617 ± 56 f		72.9 ± 0.2 b		0.271 ± 0.002 f	
		Tap water	2483 ± 146 e		58.4 ± 1.3 c		0.416 ± 0.013 e	
*S*	Unprocessed	/	5865 ± 153 a	<0.001	/	<0.001	/	<0.001
	Washing	AA	2667 ± 87 d		54.5 ± 0.4 d		0.455 ± 0.004 d	
		Na_2_CO_3_	2793 ± 140 d		52.4 ± 1.1 e		0.476 ± 0.011 c	
		NaCl	3037 ± 191 c		48.3 ± 2.0 f		0.517 ± 0.020 b	
		Tap water	3767 ± 119 b		35.8 ± 0.4 g		0.642 ± 0.004 a	
	Soaking	AA	1359 ± 57 f		76.8 ± 0.4 a		0.232 ± 0.004 g	
		Na_2_CO_3_	1434 ± 39 f		75.6 ± 0.1 a		0.244 ± 0.001 g	
		NaCl	1554 ± 101 f		73.5 ± 1.1 b		0.265 ± 0.011 f	
		Tap water	2430 ± 116 e		58.6 ± 0.9 c		0.414 ± 0.009 e	
*Rac*	Unprocessed	/	11834 ± 316 a	<0.001	/	<0.001	/	<0.001
	Washing	AA	5356 ± 179 d		54.7 ± 0.3 d		0.453 ± 0.003 d	
		Na_2_CO_3_	5637 ± 277 d		52.4 ± 1.1 e		0.476 ± 0.011 c	
		NaCl	6181 ± 397 c		47.8 ± 2.0 f		0.522 ± 0.020 b	
		Tap water	7574 ± 246 b		36.0 ± 0.4 g		0.640 ± 0.004 a	
	Soaking	AA	2741 ± 129 f		76.8 ± 0.5 a		0.232 ± 0.005 g	
		Na_2_CO_3_	2912 ± 80 f		75.4 ± 0.0 a		0.246 ± 0.000 g	
		NaCl	3171 ± 156 f		73.2 ± 0.7 b		0.268 ± 0.007 f	
		Tap water	4914 ± 262 e		58.5 ± 1.1 c		0.415 ± 0.011 e	

*P*_1_, *P*_2_, and *P*_3_ represent residue level, RR, and PF, respectively. Different lower-case letters indicate statistical significance between different treatments for mandipropamid enantiomers and racemate by Duncan’s multiple range test (*p* ≤ 0.05).

**Table 4 ijerph-19-15543-t004:** Residue level, RR, and PF values of mandipropamid in unprocessed, washing, and soaking cowpea samples.

Analyte	Treatment	Solution	Average Value ± SD (*n* = 3)					
			Residue Level (µg/kg)	*P* _1_	RR (%)	*P* _2_	PF	*P* _3_
*R*	Unprocessed	/	1528 ± 39 a	<0.001	/	<0.001	/	<0.001
	Washing	AA	702 ± 27 d		54.1 ± 0.6 e		0.459 ± 0.006 d	
		Na_2_CO_3_	900 ± 26 c		41.1 ± 0.3 f		0.589 ± 0.003 c	
		NaCl	927 ± 22 c		39.4 ± 0.2 g		0.606 ± 0.002 b	
		Tap water	1253 ± 38 b		18.0 ± 0.4 h		0.820 ± 0.004 a	
	Soaking	AA	557 ± 18 g		63.5 ± 0.3 a		0.365 ± 0.003 h	
		Na_2_CO_3_	602 ± 24 fg		60.6 ± 0.6 b		0.394 ± 0.006 g	
		NaCl	642 ± 22 ef		58.0 ± 0.4 c		0.420 ± 0.004 f	
		Tap water	669 ± 24 de		56.2 ± 0.5 d		0.438 ± 0.005 e	
*S*	Unprocessed	/	1637 ± 52 a	<0.001	/	<0.001	/	<0.001
	Washing	AA	728 ± 34 d		55.6 ± 0.7 e		0.444 ± 0.007 d	
		Na_2_CO_3_	952 ± 25 c		41.8 ± 0.3 f		0.582 ± 0.003 c	
		NaCl	982 ± 31 c		40.0 ± 0.1 g		0.600 ± 0.001 b	
		Tap water	1334 ± 47 b		18.5 ± 0.3 h		0.815 ± 0.003 a	
	Soaking	AA	580 ± 21 f		64.6 ± 0.2 a		0.354 ± 0.002 h	
		Na_2_CO_3_	629 ± 28 ef		61.6 ± 0.5 b		0.384 ± 0.005 g	
		NaCl	679 ± 27 de		58.5 ± 0.4 c		0.415 ± 0.004 f	
		Tap water	701 ± 29 d		57.2 ± 0.8 d		0.428 ± 0.008 e	
*Rac*	Unprocessed	/	3165 ± 90 a	<0.001	/	<0.001	/	<0.001
	Washing	AA	1430 ± 61 d		54.8 ± 0.7 e		0.452 ± 0.007 d	
		Na_2_CO_3_	1852 ± 50 c		41.5 ± 0.2 f		0.585 ± 0.002 c	
		NaCl	1909 ± 53 c		39.7 ± 0.1 g		0.603 ± 0.001 b	
		Tap water	2588 ± 84 b		18.3 ± 0.3 h		0.817 ± 0.003 a	
	Soaking	AA	1137 ± 39 f		64.1 ± 0.2 a		0.359 ± 0.002 h	
		Na_2_CO_3_	1231 ± 52 ef		61.1 ± 0.5 b		0.389 ± 0.005 g	
		NaCl	1321 ± 49 de		58.3 ± 0.4 c		0.417 ± 0.004 f	
		Tap water	1370 ± 53 d		56.7 ± 0.6 d		0.433 ± 0.006 e	

*P*_1_, *P*_2_, and *P*_3_ represent residue level, RR, and PF, respectively. Different lower-case letters indicate statistical significance between different treatments for mandipropamid enantiomers and racemate by Duncan’s multiple range test (*p* ≤ 0.05).

**Table 5 ijerph-19-15543-t005:** Residue level, RR, and PF values of enantiomeric and racemic mandipropamid in unprocessed and different parts of tomato and cucumber samples.

Analyte	Matrix	Average Value ± SD (*n* = 3)					
		Residue Level (µg/kg)	*P* _1_	RR (%)	*P* _2_	PF	*P* _3_
*R*	Unprocessed tomato	195 ± 5 b	<0.001	/	<0.001	/	<0.001
	Skin	661 ± 1 a		−239.8 ± 7.6		3.398 ± 0.076 a	
	Pulp	16 ± 1 c		91.6 ± 0.1 c		0.084 ± 0.001 b	
	Puree	6 ± 0 d		97.2 ± 0.0 b		0.028 ± 0.000 b	
	Juice	4 ± 0 d		97.9 ± 0.1 a		0.021 ± 0.001 b	
*S*	Unprocessed tomato	182 ± 3 b	<0.001	/	<0.001	/	<0.001
	Skin	509 ± 7 a		−180.4 ± 2.2		2.804 ± 0.022 a	
	Pulp	15 ± 0 c		91.6 ± 0.1 c		0.084 ± 0.001 b	
	Puree	5 ± 0 d		97.1 ± 0.0 b		0.029 ± 0.003 c	
	Juice	4 ± 0 d		97.9 ± 0.2 a		0.021 ± 0.002 c	
*Rac*	Unprocessed tomato	376 ± 8 b	<0.001	/	<0.001	/	<0.001
	Skin	1170 ± 8 a		−211.1 ± 4.4		3.111 ± 0.044 a	
	Pulp	32 ± 1c		91.6 ± 0.1 c		0.084 ± 0.001 b	
	Puree	11 ± 0 d		97.1 ± 0.0 b		0.029 ± 0.000 c	
	Juice	8 ± 1 d		97.9 ± 0.1 a		0.021 ± 0.001 c	
*R*	Unprocessed cucumber	605 ± 39 b	<0.001	/	< 0.001	/	<0.001
	Skin	2286 ± 91 a		−278.2 ± 18.1		3.782 ± 0.181 a	
	Pulp	6 ± 0 c		98.9 ± 0.1 c		0.011 ± 0.001 b	
	Puree	4 ± 0 c		99.4 ± 0.0 b		0.006 ± 0.000 b	
	Juice	3 ± 0 c		99.5 ± 0.0 a		0.005 ± 0.000 b	
*S*	Unprocessed cucumber	588 ± 36 b	<0.001	/	<0.001	/	<0.001
	Skin	1937 ± 66 a		−229.7 ± 10.4		3.297 ± 0.104 a	
	Pulp	6 ± 1 c		99.0 ± 0.1 b		0.010 ± 0.001 b	
	Puree	4 ± 0 c		99.4 ± 0.0 a		0.006 ± 0.000 b	
	Juice	3 ± 0 c		99.5 ± 0.0 a		0.005 ± 0.000 b	
*Rac*	Unprocessed cucumber	1194 ± 75 b	<0.001	/	<0.001	/	<0.001
	Skin	4223 ± 156 a		−254.3 ± 13.6		3.543 ± 0.13 6 a	
	Pulp	12 ± 1 c		99.0 ± 0.1 b		0.010 ± 0.001 b	
	Puree	7 ± 0 c		99.4 ± 0.0 a		0.006 ± 0.000 b	
	Juice	6 ± 1 c		99.5 ± 0.0 a		0.005 ± 0.000 b	

*P*_1_, *P*_2_, and *P*_3_ represent residue level, RR, and PF, respectively. Different lower-case letters indicate statistical significance between different treatments for mandipropamid enantiomers and racemate by Duncan’s multiple range test (*p* ≤ 0.05).

**Table 6 ijerph-19-15543-t006:** Hazard quotient (HQ) of *rac*-mandipropamid in unprocessed (UP) and processing tomato samples for different groups of Chinese consumers.

Age	Sex	b.w. (kg)	FI (g/d)	HQ (%)												
				UP	Washing				Soaking				Peeling and Juicing			
					AA	Na_2_CO_3_	NaCl	Water	AA	Na_2_CO_3_	NaCl	Water	Skin	Pulp	Puree	Juice
2–3	M	13.2	43	0.6	0.3	0.3	0.3	0.4	0.1	0.1	0.2	0.3	1.9	0.1	0.0	0.0
	F	12.3	39.6	0.6	0.3	0.3	0.3	0.4	0.1	0.1	0.2	0.3	1.9	0.1	0.0	0.0
4–6	M	16.8	56.4	0.6	0.3	0.3	0.3	0.4	0.1	0.1	0.2	0.3	2.0	0.1	0.0	0.0
	F	16.2	56.2	0.7	0.3	0.3	0.3	0.4	0.1	0.2	0.2	0.3	2.0	0.1	0.0	0.0
6–10	M	22.9	70.2	0.6	0.2	0.3	0.3	0.4	0.1	0.1	0.2	0.2	1.8	0.0	0.0	0.0
	F	21.7	65.9	0.6	0.2	0.3	0.3	0.4	0.1	0.1	0.2	0.2	1.8	0.0	0.0	0.0
11–13	M	34.1	77.2	0.4	0.2	0.2	0.2	0.3	0.1	0.1	0.1	0.2	1.3	0.0	0.0	0.0
	F	34	73.1	0.4	0.2	0.2	0.2	0.2	0.1	0.1	0.1	0.2	1.3	0.0	0.0	0.0
14–17	M	46.7	87.1	0.4	0.1	0.2	0.2	0.2	0.1	0.1	0.1	0.1	1.1	0.0	0.0	0.0
	F	45.2	81.5	0.3	0.1	0.2	0.2	0.2	0.1	0.1	0.1	0.1	1.1	0.0	0.0	0.0
18–29	M	58.4	92.1	0.3	0.1	0.1	0.2	0.2	0.1	0.1	0.1	0.1	0.9	0.0	0.0	0.0
	F	52.1	84.5	0.3	0.1	0.1	0.2	0.2	0.1	0.1	0.1	0.1	0.9	0.0	0.0	0.0
30–44	M	64.9	93.7	0.3	0.1	0.1	0.1	0.2	0.0	0.1	0.1	0.1	0.8	0.0	0.0	0.0
	F	55.7	91.3	0.3	0.1	0.1	0.2	0.2	0.1	0.1	0.1	0.1	1.0	0.0	0.0	0.0
45–59	M	63.1	99.5	0.3	0.1	0.1	0.2	0.2	0.1	0.1	0.1	0.1	0.9	0.0	0.0	0.0
	F	57	94.7	0.3	0.1	0.1	0.2	0.2	0.1	0.1	0.1	0.1	1.0	0.0	0.0	0.0
60–69	M	61.5	97.7	0.3	0.1	0.1	0.2	0.2	0.1	0.1	0.1	0.1	0.9	0.0	0.0	0.0
	F	54.3	93.2	0.3	0.1	0.1	0.2	0.2	0.1	0.1	0.1	0.1	1.0	0.0	0.0	0.0
≥70	M	58.5	88.6	0.3	0.1	0.1	0.2	0.2	0.1	0.1	0.1	0.1	0.9	0.0	0.0	0.0
	F	51	75.3	0.3	0.1	0.1	0.1	0.2	0.1	0.1	0.1	0.1	0.9	0.0	0.0	0.0

M: Male; F: Female; b.w.: Body weight; FI: Food intake; AA: acetic acid.

**Table 7 ijerph-19-15543-t007:** Hazard quotient (HQ) of *rac*-mandipropamid in unprocessed (UP) and processing cucumber samples for different groups of Chinese consumers.

Age	Sex	b.w. (kg)	FI (g/d)	HQ (%)																		
				UP	Washing				Soaking				Peeling and juicing				Pickling (d)					
					AA	Na_2_CO_3_	NaCl	Water	AA	Na_2_CO_3_	NaCl	Water	Skin	Pulp	Puree	Juice	1	3	5	7	14	21
2–3	M	13.2	43	1.9	0.9	1.0	1.0	1.5	0.5	0.6	0.6	0.8	6.9	0.0	0.0	0.0	1.5	1.1	0.9	0.8	0.8	0.7
	F	12.3	39.6	1.9	0.9	1.0	1.0	1.5	0.5	0.5	0.6	0.8	6.8	0.0	0.0	0.0	1.5	1.1	0.9	0.8	0.8	0.7
4–6	M	16.8	56.4	2.0	0.9	1.0	1.1	1.5	0.5	0.6	0.6	0.8	7.1	0.0	0.0	0.0	1.6	1.2	1.0	0.9	0.8	0.7
	F	16.2	56.2	2.1	0.9	1.1	1.1	1.6	0.5	0.6	0.6	0.8	7.3	0.0	0.0	0.0	1.6	1.2	1.0	0.9	0.8	0.7
6–10	M	22.9	70.2	1.8	0.8	1.0	1.0	1.4	0.5	0.5	0.5	0.7	6.5	0.0	0.0	0.0	1.5	1.1	0.9	0.8	0.7	0.6
	F	21.7	65.9	1.8	0.8	0.9	1.0	1.4	0.4	0.5	0.5	0.7	6.4	0.0	0.0	0.0	1.4	1.1	0.9	0.8	0.7	0.6
11–13	M	34.1	77.2	1.4	0.6	0.7	0.7	1.0	0.3	0.4	0.4	0.5	4.8	0.0	0.0	0.0	1.1	0.8	0.7	0.6	0.5	0.5
	F	34	73.1	1.3	0.6	0.7	0.7	1.0	0.3	0.4	0.4	0.5	4.5	0.0	0.0	0.0	1.0	0.7	0.6	0.6	0.5	0.5
14–17	M	46.7	87.1	1.1	0.5	0.6	0.6	0.8	0.3	0.3	0.3	0.5	3.9	0.0	0.0	0.0	0.9	0.6	0.5	0.5	0.4	0.4
	F	45.2	81.5	1.1	0.5	0.6	0.6	0.8	0.3	0.3	0.3	0.4	3.8	0.0	0.0	0.0	0.9	0.6	0.5	0.5	0.4	0.4
18–29	M	58.4	92.1	0.9	0.4	0.5	0.5	0.7	0.2	0.3	0.3	0.4	3.3	0.0	0.0	0.0	0.8	0.5	0.5	0.4	0.4	0.3
	F	52.1	84.5	1.0	0.4	0.5	0.5	0.7	0.2	0.3	0.3	0.4	3.4	0.0	0.0	0.0	0.8	0.6	0.5	0.4	0.4	0.3
30–44	M	64.9	93.7	0.9	0.4	0.4	0.5	0.7	0.2	0.2	0.3	0.3	3.0	0.0	0.0	0.0	0.7	0.5	0.4	0.4	0.3	0.3
	F	55.7	91.3	1.0	0.4	0.5	0.5	0.7	0.2	0.3	0.3	0.4	3.5	0.0	0.0	0.0	0.8	0.6	0.5	0.4	0.4	0.3
45–59	M	63.1	99.5	0.9	0.4	0.5	0.5	0.7	0.2	0.3	0.3	0.4	3.3	0.0	0.0	0.0	0.7	0.5	0.5	0.4	0.4	0.3
	F	57	94.7	1.0	0.5	0.5	0.5	0.8	0.2	0.3	0.3	0.4	3.5	0.0	0.0	0.0	0.8	0.6	0.5	0.4	0.4	0.3
60–69	M	61.5	97.7	0.9	0.4	0.5	0.5	0.7	0.2	0.3	0.3	0.4	3.4	0.0	0.0	0.0	0.8	0.5	0.5	0.4	0.4	0.3
	F	54.3	93.2	1.0	0.5	0.5	0.5	0.8	0.3	0.3	0.3	0.4	3.6	0.0	0.0	0.0	0.8	0.6	0.5	0.4	0.4	0.4
≥70	M	58.5	88.6	0.9	0.4	0.5	0.5	0.7	0.2	0.3	0.3	0.4	3.2	0.0	0.0	0.0	0.7	0.5	0.4	0.4	0.4	0.3
	F	51	75.3	0.9	0.4	0.5	0.5	0.7	0.2	0.3	0.3	0.4	3.1	0.0	0.0	0.0	0.7	0.5	0.4	0.4	0.4	0.3

M: Male; F: Female; b.w.: Body weight; FI: Food intake; AA: acetic acid.

**Table 8 ijerph-19-15543-t008:** Hazard quotient (HQ) of *rac*-mandipropamid in unprocessed (UP) and processing Chinese cabbage samples for different groups of Chinese consumers.

Age	Sex	b.w. (kg)	FI (g/d)	HQ (%)																				
				UP	Washing				Soaking				Pickling (d)						Boiling (min)			Steaming (min)		
					AA	Na_2_CO_3_	NaCl	Water	AA	Na_2_CO_3_	NaCl	Water	1	3	5	7	14	21	1	3	5	1	3	5
2–3	M	13.2	82.7	37.1	16.8	17.7	19.4	23.7	8.6	9.1	9.9	15.4	34.7	31.1	30.3	28.7	27.2	24.4	20.3	2.8	0.4	29.3	16.6	6.6
	F	12.3	82.2	39.5	17.9	18.8	20.7	25.3	9.2	9.7	10.6	16.4	37.0	33.2	32.3	30.6	29.0	26.0	21.6	3.0	0.4	31.3	17.7	7.1
4–6	M	16.8	105.8	37.3	16.9	17.7	19.5	23.8	8.6	9.2	10.0	15.5	34.9	31.2	30.4	28.9	27.4	24.5	20.4	2.8	0.4	29.5	16.7	6.7
	F	16.2	101.8	37.2	16.8	17.7	19.4	23.8	8.6	9.1	10.0	15.4	34.8	31.2	30.4	28.8	27.3	24.4	20.3	2.8	0.4	29.4	16.7	6.6
6–10	M	22.9	137.5	35.5	16.1	16.9	18.6	22.7	8.2	8.7	9.5	14.8	33.3	29.8	29.0	27.5	26.1	23.4	19.4	2.7	0.4	28.1	15.9	6.3
	F	21.7	132.6	36.2	16.4	17.2	18.9	23.1	8.4	8.9	9.7	15.0	33.9	30.3	29.5	28.0	26.5	23.8	19.8	2.7	0.4	28.6	16.2	6.5
11–13	M	34.1	156.6	27.2	12.3	12.9	14.2	17.4	6.3	6.7	7.3	11.3	25.5	22.8	22.2	21.1	19.9	17.9	14.9	2.0	0.3	21.5	12.2	4.9
	F	34	155.3	27.0	12.2	12.9	14.1	17.3	6.3	6.7	7.2	11.2	25.3	22.7	22.1	20.9	19.8	17.8	14.8	2.0	0.3	21.4	12.1	4.8
14–17	M	46.7	178.9	22.7	10.3	10.8	11.8	14.5	5.3	5.6	6.1	9.4	21.2	19.0	18.5	17.6	16.6	14.9	12.4	1.7	0.2	17.9	10.2	4.1
	F	45.2	156	20.4	9.2	9.7	10.7	13.1	4.7	5.0	5.5	8.5	19.1	17.1	16.7	15.8	15.0	13.4	11.2	1.5	0.2	16.2	9.1	3.6
18–29	M	58.4	202.2	20.5	9.3	9.8	10.7	13.1	4.7	5.0	5.5	8.5	19.2	17.2	16.7	15.9	15.0	13.5	11.2	1.5	0.2	16.2	9.2	3.7
	F	52.1	186.2	21.1	9.6	10.1	11.0	13.5	4.9	5.2	5.7	8.8	19.8	17.7	17.3	16.4	15.5	13.9	11.6	1.6	0.2	16.7	9.5	3.8
30–44	M	64.9	206.5	18.8	8.5	9.0	9.8	12.0	4.4	4.6	5.0	7.8	17.6	15.8	15.4	14.6	13.8	12.4	10.3	1.4	0.2	14.9	8.4	3.4
	F	55.7	192.8	20.5	9.3	9.8	10.7	13.1	4.7	5.0	5.5	8.5	19.2	17.2	16.7	15.9	15.0	13.5	11.2	1.5	0.2	16.2	9.2	3.7
45–59	M	63.1	211.4	19.8	9.0	9.4	10.4	12.7	4.6	4.9	5.3	8.2	18.6	16.6	16.2	15.4	14.6	13.0	10.8	1.5	0.2	15.7	8.9	3.5
	F	57	194.9	20.2	9.2	9.6	10.6	12.9	4.7	5.0	5.4	8.4	19.0	17.0	16.5	15.7	14.9	13.3	11.1	1.5	0.2	16.0	9.1	3.6
60–69	M	61.5	187.7	18.1	8.2	8.6	9.4	11.6	4.2	4.4	4.8	7.5	16.9	15.1	14.8	14.0	13.3	11.9	9.9	1.4	0.2	14.3	8.1	3.2
	F	54.3	170.8	18.6	8.4	8.9	9.7	11.9	4.3	4.6	5.0	7.7	17.4	15.6	15.2	14.4	13.7	12.2	10.2	1.4	0.2	14.7	8.3	3.3
≥70	M	58.5	172.2	17.4	7.9	8.3	9.1	11.1	4.0	4.3	4.7	7.2	16.3	14.6	14.2	13.5	12.8	11.5	9.5	1.3	0.2	13.8	7.8	3.1
	F	51	151.7	17.6	8.0	8.4	9.2	11.3	4.1	4.3	4.7	7.3	16.5	14.8	14.4	13.6	12.9	11.6	9.6	1.3	0.2	13.9	7.9	3.1

M: Male; F: Female; b.w.: Body weight; FI: Food intake; AA: acetic acid.

**Table 9 ijerph-19-15543-t009:** Hazard quotient (HQ) of *rac*-mandipropamid in unprocessed (UP) and processing cowpea samples for different groups of Chinese consumers.

Age	Sex	b.w.(kg)	FI (g/d)	HQ (%)																				
				UP	Washing				Soaking				Pickling (d)						Boiling (min)			Steaming (min)		
					AA	Na_2_CO_3_	NaCl	Water	AA	Na_2_CO_3_	NaCl	Water	1	3	5	7	14	21	1	3	5	1	3	5
2–3	M	13.2	82.7	9.9	4.5	5.8	6.0	8.1	3.6	3.9	4.1	4.3	8.7	7.6	7.0	6.1	5.2	4.7	3.3	0.4	0.0	4.3	0.5	0.0
	F	12.3	82.2	10.6	4.8	6.2	6.4	8.6	3.8	4.1	4.4	4.6	9.3	8.1	7.5	6.5	5.5	5.0	3.6	0.5	0.0	4.6	0.6	0.1
4–6	M	16.8	105.8	10.0	4.5	5.8	6.0	8.1	3.6	3.9	4.2	4.3	8.8	7.7	7.1	6.1	5.2	4.7	3.3	0.4	0.0	4.4	0.5	0.0
	F	16.2	101.8	9.9	4.5	5.8	6.0	8.1	3.6	3.9	4.1	4.3	8.7	7.6	7.1	6.1	5.2	4.7	3.3	0.4	0.0	4.3	0.5	0.0
6–10	M	22.9	137.5	9.5	4.3	5.6	5.7	7.8	3.4	3.7	4.0	4.1	8.4	7.3	6.8	5.8	5.0	4.5	3.2	0.4	0.0	4.2	0.5	0.0
	F	21.7	132.6	9.7	4.4	5.7	5.8	7.9	3.5	3.8	4.0	4.2	8.5	7.4	6.9	5.9	5.1	4.6	3.2	0.4	0.0	4.2	0.5	0.0
11–13	M	34.1	156.6	7.3	3.3	4.3	4.4	5.9	2.6	2.8	3.0	3.1	6.4	5.6	5.2	4.5	3.8	3.4	2.4	0.3	0.0	3.2	0.4	0.0
	F	34	155.3	7.2	3.3	4.2	4.4	5.9	2.6	2.8	3.0	3.1	6.4	5.6	5.1	4.4	3.8	3.4	2.4	0.3	0.0	3.2	0.4	0.0
14–17	M	46.7	178.9	6.1	2.7	3.5	3.7	5.0	2.2	2.4	2.5	2.6	5.3	4.7	4.3	3.7	3.2	2.9	2.0	0.3	0.0	2.7	0.3	0.0
	F	45.2	156	5.5	2.5	3.2	3.3	4.5	2.0	2.1	2.3	2.4	4.8	4.2	3.9	3.4	2.9	2.6	1.8	0.2	0.0	2.4	0.3	0.0
18–29	M	58.4	202.2	5.5	2.5	3.2	3.3	4.5	2.0	2.1	2.3	2.4	4.8	4.2	3.9	3.4	2.9	2.6	1.8	0.2	0.0	2.4	0.3	0.0
	F	52.1	186.2	5.7	2.6	3.3	3.4	4.6	2.0	2.2	2.4	2.4	5.0	4.3	4.0	3.5	3.0	2.7	1.9	0.3	0.0	2.5	0.3	0.0
30–44	M	64.9	206.5	5.0	2.3	2.9	3.0	4.1	1.8	2.0	2.1	2.2	4.4	3.9	3.6	3.1	2.6	2.4	1.7	0.2	0.0	2.2	0.3	0.0
	F	55.7	192.8	5.5	2.5	3.2	3.3	4.5	2.0	2.1	2.3	2.4	4.8	4.2	3.9	3.4	2.9	2.6	1.8	0.2	0.0	2.4	0.3	0.0
45–59	M	63.1	211.4	5.3	2.4	3.1	3.2	4.3	1.9	2.1	2.2	2.3	4.7	4.1	3.8	3.3	2.8	2.5	1.8	0.2	0.0	2.3	0.3	0.0
	F	57	194.9	5.4	2.4	3.2	3.3	4.4	1.9	2.1	2.3	2.3	4.8	4.2	3.8	3.3	2.8	2.6	1.8	0.2	0.0	2.4	0.3	0.0
60–69	M	61.5	187.7	4.8	2.2	2.8	2.9	3.9	1.7	1.9	2.0	2.1	4.2	3.7	3.4	3.0	2.5	2.3	1.6	0.2	0.0	2.1	0.3	0.0
	F	54.3	170.8	5.0	2.2	2.9	3.0	4.1	1.8	1.9	2.1	2.2	4.4	3.8	3.5	3.1	2.6	2.4	1.7	0.2	0.0	2.2	0.3	0.0
≥70	M	58.5	172.2	4.7	2.1	2.7	2.8	3.8	1.7	1.8	1.9	2.0	4.1	3.6	3.3	2.9	2.4	2.2	1.6	0.2	0.0	2.0	0.3	0.0
	F	51	151.7	4.7	2.1	2.8	2.8	3.8	1.7	1.8	2.0	2.0	4.1	3.6	3.3	2.9	2.5	2.2	1.6	0.2	0.0	2.1	0.3	0.0

M: Male; F: Female; b.w.: Body weight; FI: Food intake; AA: acetic acid.

## Data Availability

Data is contained within the article and Appendix A.

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
