# Peer review of "Effects of Household Processing on Residues of the Chiral Fungicide Mandipropamid in Four Common Vegetables"

_ijerph, 2022, doi:10.3390/ijerph192315543_

Round 1

Reviewer 1 Report (New Reviewer)

The manuscript ID ijerph-2017260 entitled “Effects of Household Processing on Residues of the Chiral Fungicide Mandipropamid in Four Common Vegetables" attempts to study the residual content of mandipropamid in unprocessed and processed vegetables and assess the health risks associated with their intake. In general, this is an interesting research that has important health implications, especially accounting for cumulative exposures; consequently, the work is worthy of publication in the International Journal of Environmental Research and Public Health. However, several irregularities in the paper preclude acceptance in its present form. The following shortcomings stand out in the presented manuscript:

L9. According to IUPAC terminology, “concentration” characterizes the composition of a mixture with respect to the volume; “content” represents the amount of a component divided by the system´s mass.

L51. octanol/water distribution coefficient!

L58-60. The meaning of the sentence is not clear.

L66. Removing, degrading or inactivating?

L100. Where did you source the chemical?

L100-101. 140.4 g a.i./hm2 (the recommended high dosage) three times with an interval of 7 d.… The dosage and the number of applications of the fungicide lacked theoretic support. The label of Revus® stipulates a maximum of 4 applications per year, with no more than 2 consecutive applications when applied alone and a minimum spray interval of 10 d. A maximum cumulative application rate of 900 g ac/ha is recommended, with 4 applications at successive 14 d intervals.

L102. Small pieces… average size?

L104. Three replicates. Establishing statistical significances with merely three replicates per sample is questionable, a minimum of five replicates would be more appropriate.

L110-111. Please delete these lines. Washing, an initial procedure in household preparation, is one of the most common household processing approaches [17].

L111-114. 10% (v/v), 10% (w/v)…

L115-116. It would be helpful to know what time of the year the samples were obtained, and if they were processed right-a-way, or maintained somehow.

L116-117. Please delete these lines. Soaking is a long-term washing method that removes pesticide residue from vegetables more effectively.

L122-125. Please delete these lines. Pickled vegetables refer to processed products after infiltrating salt into vegetable tissues, selectively regulating microbial fermentation [20]. Spontaneous fermentation without the use of starter cultures or sterilization is popular in Chinese household vegetable pickling [21].

L126. Same as L102.

L132-133. Please delete these lines. Peeling and juicing are two critical steps in the household processing of tomato and cucumber [5,24].

L133-134. The authors should consider the use of knife tools with controlled depth of knife penetration.

L134-136. The sentence is somewhat confusing.

L138.39. Please delete these lines. Boiling and steaming are two common cooking methods during the household processing procedure [26].

L140. were boiled in 100°C? At sea level, water boils at 100° C. At higher altitudes, the temperature of the boiling point is lower.

L141. steamed over boiling water… please add the temperature of the steam!

L167-168. How was the daily consumption of the plants calculated? What references were used to obtain these values?

L174. The Duncan´s multiple range test is a widely criticized statistical post-hoc test, because it has much less control over the type I error rate.

L177. Please improve the analysis and interpretation of results.

L184-187. How was it determined?

L194-196. How was it determined?

L200-203. This is not clear.

Tables 1-5 and Figures 2 and 3. You should insert the moisture content of all evaluated samples in the corresponding table or figure. Otherwise, the levels of pesticide residue content need to be expressed as μg/kg dry weight.

Tables 1 and 2. Please recheck the racemate residue values!

L210, 215,220, 225 and 267. P1, P2 and P3 represent residue, RR and PF, respectively. Residue? Please clarify. p ≤ 0.05!

L228. Pickling

L234-235. The meaning of the sentence is not clear.

L239-241. So, what happened at 14 and 21 d?

L251-254. This is not clear.

L280-285.  It is not clear to the reader what this sentence means.

Figure 4. Statistical analyses should be performed and reported for all parameters (EF values).

L295. Tables S3-S6. Why were results considered as supplementary material? These tables should be part of the manuscript.

L310. the removal efficiencies of washing in AA solution… Please add the concentration of AA, Na2CO3, and NaCl.

L312. for eight pesticides… Does Kin and Huat/Soliman/Zohair discuss the removal of mandipropamid? This needs to be explained somewhere.

L314-317. Did the authors evaluate the concentration of mandipropamid in the soaking/washing water? how to face the problem of fungicide residues in water?

L339-342. It is well known that mandipropamid is rapidly adsorbed to the wax layer of the plant surfaces!

L359. According to the MRL values, in this study, Chinese cabbage is safe for consumption without prior household treatment.

L365-366. and the cowpea sample was safe after cooking… According to Figure 3, boiling and steaming for 1 min were not effective!

L366. Does the cumulative pesticide intake, when the HQchronic is less than 100% consider any interactions between the multiple food consumed?

L371. More emphasis on finding and its implication may be mentioned in the conclusion section.

Author Response

Dear reviewer,

On behalf of all co-authors, I would like to thank you very much for the constructive comments on our manuscript. We have modified our manuscript following your precious recommendations and addressed every comment to the best of our knowledge. The changes are highlighted in red in the revised manuscript. We herein submit the revised manuscript to be considered for publication. Thank you for your helpful coordination and understanding throughout the evaluation process of our manuscript. All Changes Refer to the Previously Submitted Version of the Manuscript.

Reviewers' comments:

Reviewer 1#:

The manuscript ID ijerph-2017260 entitled “Effects of Household Processing on Residues of the Chiral Fungicide Mandipropamid in Four Common Vegetables" attempts to study the residual content of mandipropamid in unprocessed and processed vegetables and assess the health risks associated with their intake. In general, this is an interesting research that has important health implications, especially accounting for cumulative exposures; consequently, the work is worthy of publication in the International Journal of Environmental Research and Public Health. However, several irregularities in the paper preclude acceptance in its present form. The following shortcomings stand out in the presented manuscript:

  1. L9. According to IUPAC terminology, “concentration” characterizes the composition of a mixture with respect to the volume; “content” represents the amount of a component divided by the system´s mass.

Answer: Thank you for your suggestion. On line 9 of page 1 in the previous paper, “concentration” was changed to “content”.

  1. L51. octanol/water distribution coefficient!

 Answer: Thank you for your suggestion. On line 51 of page 2 in the previous paper, “water-octanol distribution coefficient” was changed to “octanol/water distribution coefficient”.

  1. L58-60. The meaning of the sentence is not clear.

 Answer: Thank you for your suggestion. We have rewritten this sentence. On line 58-60 of page 2 in the previous paper, “Mandipropamid, which can effective prevention from most foliar oomycete pathogens, is a chiral systemic fungicide with a broad bactericidal spectrum, and its market sales have been growing since the last half decade [13].” was changed to “Mandipropamid is a chiral systemic fungicide with a broad bactericidal spectrum and can effective prevention from most foliar oomycete pathogens. The market sales of mandipropamid have been growing since the last half decade [13].”.

  1. L66. Removing, degrading or inactivating?

Answer: Thank you for your suggestion. The main aim of this paper is to investigate the effect of different household processing methods in reducing the amount of mandipropamid in contaminated vegetable samples. Therefore, we used the word “removing” to present the research aim.

  1. L100. Where did you source the chemical?

Answer: Thank you for your suggestion. The formulation of mandipropamid suspension concentrate (SC) (23.4%) was purchased from Syngenta Nantong Crop Protection Co., Ltd. (Nantong, China), which has been mentioned in “2.1. Chemicals and reagents” section.

  1. L100-101. 140.4 g a.i./hm2(the recommended high dosage) three times with an interval of 7 d.… The dosage and the number of applications of the fungicide lacked theoretic support. The label of Revus® stipulates a maximum of 4 applications per year, with no more than 2 consecutive applications when applied alone and a minimum spray interval of 10 d. A maximum cumulative application rate of 900 g ac/ha is recommended, with 4 applications at successive 14 d intervals.

Answer: Thank you for your suggestion. The application doss, spraying time, and interval are obtained from the label of 23.4% mandipropamid SC (Syngenta Nantong Crop Protection Co., Ltd.). For the prevention of late blight, an application dose of 105.3-140.4 g a.i./hm2 was used, with two and three application times, and an interval of 7-10 d. Therefore, we used the highest application dose, most application time, and shortest interval to present the worst conditions.

  1. L102. Small pieces… average size?

Answer: Thank you for your suggestion. Before pretreatment, vegetable samples were cut into approximately one cubic centimeter to facilitate homogenization. On line 101-103 of page 3 in the previous paper, “After different processing procedures, the vegetable samples were chopped into small pieces and then homogenized before pre-treatment.” was changed to “After different processing procedures, the vegetable samples were chopped into small pieces with an average size of approximately 1 cm3 and then homogenized before pre-treatment.”.

  1. L104. Three replicates. Establishing statistical significances with merely three replicates per sample is questionable, a minimum of five replicates would be more appropriate.

Answer: Thank you for your suggestion. Based on the previous references, each treatment was repeated three times with the same technical operation, and the experimental operation procedure was rigorously followed within the allowable errors to ensure that the experimental results were accurate and reliable. However, in the future studies we will perform the trials based on the suggestions of the reviewer.

Reference:

  1. Bonnechère, A.; Hanot, V.; Bragard, C.; Bedoret, T.; van Loco, J. (2012). Effect of household and industrial processing on the levels of pesticide residues and degradation products in melons. Food Addit. Contam. A. 2012, 29, 1058–1066. https://org/10.1016/j.foodcont.2011.11.010.
  2. Polat, B.; Tiryaki, O. Assessing washing methods for reduction of pesticide residues in Capia pepper with LC-MS/MS. Environ. Sci. Heal. B. 2020, 55, 1–10. https://doi.org/10.1080/03601234.2019.1660563.
  3. Liu, N.; Dong, F.S.; Liu, X.G.; Xu, J.; Li, Y.B.; Han, Y.T.; Zhu, Y.L.; Cheng, Y.P.; Chen, Z.L.; Tao, Y.; Zheng, Y.Q. Effect of household canning on the distribution and reduction of thiophanate-methyl and its metabolite carbendazim residues in tomato. Food Control. 2014, 43, 115–120. http://dx.doi.org/10.1016/j.foodcont.2014.03.003.

  1. L110-111. Please delete these lines. Washing, an initial procedure in household preparation, is one of the most common household processing approaches [17].

Answer: Thank you for your suggestion. On line 110-111 of 3 in the previous paper, “Washing, an initial procedure in household preparation, is one of the most common household processing approaches [17].” was deleted.

  1. L111-114. 10% (v/v), 10% (w/v)…

Answer: Thank you for your suggestion. On line 111-114 of page 3 in the previous paper, “Four kinds of non-toxic solutions, including 10% AA solution (acidic solution), 10% NaCl solution (neutral solution), 10% Na2CO3 solution (alkaline solution) and tap water, were chosen to assess the effect of washing on the residues mandipropamid enantiomers in tomato, cucumber, Chinese cabbage, and cowpea.” was changed to “Four kinds of non-toxic solutions, including 10% (v/v) AA solution (acidic solution), 10% (w/v) NaCl solution (neutral solution), 10% Na2CO3 (w/v) solution (alkaline solution) and tap water, were chosen to assess the effect of washing on the residues mandipropamid enantiomers in tomato, cucumber, Chinese cabbage, and cowpea.”.

  1. L115-116. It would be helpful to know what time of the year the samples were obtained, and if they were processed right-a-way, or maintained somehow.

Answer: Thank you for your suggestion. We have added some mentioned information. On line 115-116 of page 3 in the previous paper, “Mandipropamid-treated vegetable samples collected in the field trials were washed in different solutions for 1 min [18].” was changed to “Mandipropamid-treated vegetable samples collected in the field trials at August 2021, which were directly transferred to the lab and prepared as lab samples, were washed in different solutions for 1 min [17, 18].”.

12.L116-117. Please delete these lines. Soaking is a long-term washing method that removes pesticide residue from vegetables more effectively.

Answer: Thank you for your suggestion. On line 116-117 of page 3 in the previous paper, “Soaking is a long-term washing method that removes pesticide residue from vegetables more effectively.” was deleted.

13.L122-125. Please delete these lines. Pickled vegetables refer to processed products after infiltrating salt into vegetable tissues, selectively regulating microbial fermentation [20]. Spontaneous fermentation without the use of starter cultures or sterilization is popular in Chinese household vegetable pickling [21].

Answer: Thank you for your suggestion. On line 122-15 of page 3-4 in the previous paper, “Pickled vegetables refer to processed products after infiltrating salt into vegetable tissues, selectively regulating microbial fermentation [20]. Spontaneous fermentation without the use of starter cultures or sterilization is popular in Chinese household vegetable pickling [21].” was deleted.

13.L126. Same as L102.

Answer: Thank you for your suggestion. On line 125-127 of page 4 in the previous paper, “The mandipropamid-contaminanted cucumber samples were cut into small pieces, and the mandipropamid-contaminated Chinese cabbage and cowpea samples were drained as the raw material of sauerkraut [22,23].” was changed to “The contaminated cucumber samples were cut into small pieces with an average size of approximately 1 cm3, and the contaminated Chinese cabbage and cowpea samples were drained as the raw material of sauerkraut [20-23].”.

14.L132-133. Please delete these lines. Peeling and juicing are two critical steps in the household processing of tomato and cucumber [5,24].

Answer: Thank you for your suggestion. On line 132-133 of page 4 in the previous paper, “Peeling and juicing are two critical steps in the household processing of tomato and cucumber [5,24].” was deleted.

15.L133-134. The authors should consider the use of knife tools with controlled depth of knife penetration.

Answer: Thank you for your suggestion. The tools used in this paper are mainly used to remove the skin from vegetable samples. The thickness of the tool used is 1 mm, and the thickness of the peel is controlled to be about 1 mm.

16.L134-136. The sentence is somewhat confusing.

Answer: Thank you for your suggestion. On line 134-136 of page 4 in the previous paper, “One part of the pulp sample was then detected and the remaining part homogenized to obtain the juice and puree samples.” was changed to “The peel sample and a partial pulp sample were subsequently detected. Another part of the pulp sample was homogenized to obtain the juice and puree samples.”.

17.L138-139. Please delete these lines. Boiling and steaming are two common cooking methods during the household processing procedure [26].

Answer: Thank you for your suggestion. On line 138-139 of page 4 in the previous paper, “Boiling and steaming are two common cooking methods during the household processing procedure [26].” was deleted.

18.L140. were boiled in 100°C? At sea level, water boils at 100°C. At higher altitudes, the temperature of the boiling point is lower.

Answer: Thank you for your suggestion. The boiling point of water was 95°C in our lab. Therefore, we have made corrections for this error. Figure 1 has been redrawn and on line 139-141 of page 4 in the previous paper, “Part of the mandipropamid-contaminated Chinese cabbage or cowpea samples were boiled in 100 °C water for 1, 3, and 5 min, and another part was steamed over boiling water for 1, 3, and 5 min [27].” was changed to “Part of the mandipropamid-contaminated Chinese cabbage or cowpea samples were boiled in 93 °C water for 1, 3, and 5 min, and another part was steamed over boiling water (96 °C) for 1, 3, and 5 min [26, 27].”.

19.L141. steamed over boiling water… please add the temperature of the steam!

Answer: Thank you for your suggestion. We have added the temperature. On line 139-141 of page 4 in the previous paper, “Part of the mandipropamid-contaminated Chinese cabbage or cowpea samples were boiled in 100 °C water for 1, 3, and 5 min, and another part was steamed over boiling water for 1, 3, and 5 min [27].” was changed to “Part of the mandipropamid-contaminated Chinese cabbage or cowpea samples were boiled in 95 °C water for 1, 3, and 5 min, and another part was steamed over boiling water (95 °C) for 1, 3, and 5 min [26, 27].”.

20.L167-168. How was the daily consumption of the plants calculated? What references were used to obtain these values?

Answer: Thank you for your suggestion. Dietary intake of pesticides is primarily determined by the concentration of pesticide residues in the food and the average food intake per capita. In this study, daily intake was calculated based on previous studies and dietary data in combination with mandipropamid. When evaluating the risk of chronic toxicity associated with the dietary intake of pesticides, the international estimated daily intake (IEDI, mg kg–1, bw) is compared with the acceptable daily intake (ADI, mg kg–1, bw) to determine whether the risk is acceptable. The IEDI value is obtained from the equation IEDI = (FI×RL)/b.w., where FI is the food intake data, RL is the residue level of mandipropamid in processed food samples, and b.w. is the body weight (about 60kg). The ADI value for mandipropamid is 0.2 mg/kg b.w.

References:

  1. JMPR (Joint FAO/WHO Meeting on Pesticide Residues). JMPR evaluation: mandipropamid. http://www.fao.org/fileadmin/templates/agphome/documents/PestsPesticides/JMPR/Evaluation08/Mandipropamid.pdf. (accessed 10 February 2022).
  2. Wang, W.Z.; Teng, P.P.; Liu, F.M.; Fan, T.T.; Peng, Q.R.; Wang, Z.Y.; Hou, T.Y. Residue analysis and risk assessment of oxathiapiprolin and its metabolites in cucumbers under field conditions. Agric. Food Chem. 2019, 67, 12904-12910.

21.L174. The Duncan´s multiple range test is a widely criticized statistical post-hoc test, because it has much less control over the type I error rate.

Answer: Thank you for your suggestion. The scope of application of Duncan 's multiple interval test is to compare the difference between two or more sample wood means. In this case, the t-test method cannot be used for a pairwise comparison, which necessarily increases the possibility of two types of errors. Therefore, methods of variance analysis must be used to compare the means of more than two groups. Of course, the method of ANOVA is also applicable to the comparison of the mean values of the two groups. The variational analysis can be completed by invoking this procedure. This process can only be done with a one-way analysis of variance, that is, with a completely random design data analysis of variance. The application in this study is exactly as required, and the control of type I error rate is tightly controlled.

22.L177. Please improve the analysis and interpretation of results.

Answer: Thank you for your suggestion. We have made some revisions to improve the interpretation of results, which were shown in the revised paper.

23.L184-187. How was it determined?

Answer: Thank you for your suggestion. Average recovery was added to evaluate the accuracy and precision of the mandipropamid residue detection method on tomatoes, cucumbers, cowpeas, and cabbages. At the addition levels of 2.5-500 μg/kg, five replicate experiments were performed for each sample. The average recovery rate was between 70 % and 120 %. The smaller the RSD value, the higher the recovery rate of mandipropamid in the four vegetable samples extracted by the residue detection method. The accuracy and precision of the proposed method meet the requirements of pesticide residue analysis.

24.L194-196. How was it determined?

Answer: Thank you for your suggestion. Cucumber samples were washed with different solutions (10% NaCl, 10% Na2CO3, and 10% AA). The data for mandipropamid enantiomers and racemate residues are shown in Table 2. The RR is calculated from the residues to compare different washing treatments.

25.L200-203. This is not clear.

Answer: Thank you for your suggestion. We have rewritten the sentence. On line 200-203 of page 5 in the previous paper, “In Table 1 and Table 2, the RR values of the three analytes in tomato and cucumber samples soaked in AA, Na2CO3, NaCl solution and tap water were 75.3-82.2%, 71.5-77.0%, 69.9-73.1% and 57.8-60.1%, respectively, which were significantly higher than those in samples washed in the four solutions (P < 0.001).” was changed to “In Table 1 and Table 2, the RR values of the three analytes in tomato and cucumber samples soaked in 10% AA solution (75.3-82.2%), 10% Na2CO3 solution (71.5-77.0%), 10% NaCl solution (69.9-73.1%) and tap water (57.8-60.1%) were significantly higher than those in samples washed in the four solutions (P < 0.001).”.

26.Tables 1-5 and Figures 2 and 3. You should insert the moisture content of all evaluated samples in the corresponding table or figure. Otherwise, the levels of pesticide residue content need to be expressed as μg/kg dry weight.

Answer: Thank you for your suggestion. In our study, the content of mandipropamid has been detected in fresh or treated vegetable samples. Moisture content may vary in different raw or treated samples, however, the concentration of mandipropamid could be useful to evaluate the removal efficiency of a particular processing method, regardless of the moisture content. Therefore, the moisture content was not taken into account during the processing trials.

27.Tables 1 and 2. Please recheck the racemate residue values!

Answer: Thank you for your suggestion. We have check the rac-mandipropamid residual data and made some corrections.

28.L210, 215,220, 225 and 267. P1, P2 and P3 represent residue, RR and PF, respectively. Residue? Please clarify. p ≤ 0.05!

Answer: Thank you for your suggestion. We have made some revisions, such as changing “residue” to “residue level”, change “P = 0.05” to “P ≤ 0.05”.

29.L228. Pickling

Answer: Thank you for your suggestion. On line 228 of page 9 in the previous paper, “picking” was changed to “Pickling”.

30.L234-235. The meaning of the sentence is not clear.

Answer: Thank you for your suggestion. On line 234-235 of page 9 in the previous paper, “The PF data (Figure 2) are also demonstrated by different RR values for mandipropamid in different vegetable samples after pickling.” was changed to “The PF data (Figure 2) are also demonstrated the different removal efficiencies for mandipropamid in different vegetable samples after pickling.”.

31.L239-241. So, what happened at 14 and 21 d?

Answer: Thank you for your suggestion. The concentration of mandipropamid is reduced due to biodegradation. On line 239-241 of page 9 in the previous paper, “In Table S2 (Supporting information), the concentrations of mandipropamid enantiomers and racemate in brine samples increased from the first day to the seventh day.” was changed to “In Table S2 (Supporting information), the concentrations of mandipropamid enantiomers and racemate in brine samples increased from the first day to the seventh day and then decreased in the following 14 days”.

32.L251-254. This is not clear.

Answer: Thank you for your suggestion. We have rewritten this sentence. On line 251-254 of page 10 in the previous paper, “The respective RR values of mandipropamid residues in puree and juice samples are 97.1-97.2% (PF = 0.028-0.029) and 97.9% (PF = 0.021) for tomato and 99.4% (PF = 0.006) and 99.5% (PF = 0.005) for cucumber.” was changed to “The RR values of mandipropamid residues in puree and juice samples are 97.1- 97.9% for tomato and 99.4-99.5% for cucumber. The corresponding PF values are in the range of 0.005-0.029.”.

33.L280-285.  It is not clear to the reader what this sentence means.

Answer: Thank you for your suggestion. We have rewritten this part. On line 280-285 of page 11-12 in the previous paper, “Similar results were also found in for pickled vegetable samples, which showed no significant enantioselectivity in the washing, soaking, and pickling removal efficiency for mandipropamid in contaminated vegetable samples. Although after peeling or cooking, a few EF values were changed considerably, such as in the skin samples (0.565 and 0.541) or cooking for 5 min in cowpea samples (0.447 and 0.382, Figure 4), the reduction of mandipropamid enantiomers was not different in most cases.” was changed to “No significant enantioselectivity was also found in the pickled vegetable samples. Although after peeling or cooking, a few EF values were changed considerably, such as EF values increased to 0.565 in tomato skin samples and to 0.541 in cucumber skin samples or EF values decreased to 0.447 after steaming for 5 min and to 0.382 after boiling for 5 min in cowpea samples (Figure 4), the reduction of mandipropamid enantiomers was not different in most cases.”.

34.Figure 4. Statistical analyses should be performed and reported for all parameters (EF values).

Answer: Thank you for your suggestion. We have added the results of statistical analyses in Figure 4.

35.L295. Tables S3-S6. Why were results considered as supplementary material? These tables should be part of the manuscript.

Answer: Thank you for your suggestion. Tables S3-S6 in the supplementary file have been moved in the manuscript and renamed as Tables 6-8.

36.L310. the removal efficiencies of washing in AA solution… Please add the concentration of AA, Na2CO3, and NaCl.

Answer: Thank you for your suggestion. On line 309-312 of page 12-13 in the previous paper, “For example, Kin and Huat found that for cucumber and strawberry samples, the removal efficiencies of washing in AA solution (RR values of 44-70%) were more excellent than those in Na2CO3 solution (RR values of 30-50%), NaCl solution (RR values of 23-40%) and tap water (RR values of 10-20%) for eight pesticides [18].” was changed to “For example, Kin and Huat found that for cucumber and strawberry samples, the removal efficiencies of washing in 10% AA solution (RR values of 44-70%) were more excellent than those in 10% Na2CO3 solution (RR values of 30-50%), 10% NaCl solution (RR values of 23-40%) and tap water (RR values of 10-20%) for eight pesticides [18].”.

37.L312. for eight pesticides… Does Kin and Huat/Soliman/Zohair discuss the removal of mandipropamid? This needs to be explained somewhere.

Answer: Thank you for your suggestion. The eight pesticides reported by Kin and Huat/Soliman/Zohair were not included in mandipropamid. The purpose of citing this reference is to illustrate the better removal efficiency found by the 10% AA solution compared to other washing solutions. We have made some revisions. On line 308-309 of page 12 in the previous paper, “Similar results have been reported in a number of previous studies.” was changed to “Similar results have been reported for other pesticides in a number of previous studies.”.

38.L314-317. Did the authors evaluate the concentration of mandipropamid in the soaking/washing water? how to face the problem of fungicide residues in water?

Answer: Thank you for your suggestion. Concentrations of mandipropamid have been detected in soaking and rinsing water. The residue levels were very low (< the lowest spiked levels), which could be negligible for causing environmental risk.

39.L339-342. It is well known that mandipropamid is rapidly adsorbed to the wax layer of the plant surfaces!

Answer: Thank you for your suggestion. Mandipropamid has a high affinity for the wax layer on the leaf surface and can be quickly absorbed by the leaf and remain in the wax layer on the leaf surface, protecting the leaf from the disease. However, mandipropamid has a poor solubility in water, and the small amount of it absorbed by leaf remains in the waxy layer, which weakens internal transport. Therefore, peeling better removes mandipropamid content.

40.L359. According to the MRL values, in this study, Chinese cabbage is safe for consumption without prior household treatment.

Answer: Thank you for your suggestion. No formal MRL of mandipropamid in Chinese cabbage has been established by any country or organization. 25 mg/kg is only a temporary MRL value proposed by the Chinese authorities. While this MRL may ensure the safety of almost all contaminated Chinese cabbage samples, the need for residue studies should be noted and this study could provide some data basis for establishing a quantitative limit standard for mandipropamid in Chinese cabbage.

41.L365-366. and the cowpea sample was safe after cooking… According to Figure 3, boiling and steaming for 1 min were not effective!

Answer: Thank you for your suggestion. We have made some changes to clarify this point. On line 365-366 of page 14 in the previous paper, “and the cowpea sample was safe after cooking” was changed to “and the cowpea sample was safe after cooking (> 1 min)”.

42.L366. Does the cumulative pesticide intake, when the HQchronic is less than 100% consider any interactions between the multiple food consumed?

Answer: Thank you for your suggestion. In this study, four vegetable samples, cucumber, cowpea, Chinese cabbage, and tomato, were tested independently, so interactions between multiple samples were not considered.

43.L371. More emphasis on finding and its implication may be mentioned in the conclusion section

Answer: Thank you for your suggestion. The “Conclusion” section was changed to “In this study, the effects of different processing techniques, including washing, soaking, pickling, peeling, juicing, boiling, and steaming, on mandipropamid residues were investigated in four types of vegetables. The results showed that the concentration of mandipropamid decreased after the above processing procedure. The addition of 10% AA, 10% Na2CO3, and 10% NaCl significantly increased the removal efficiency of mandipropamid compared to tap water washing and soaking. The PF values (0.178-0.459) demonstrated that washing and soaking with the acidic solution reduced the mandipropamid residue more than the other three solutions. The decrease in PF from 0.938 to 0.348 illustrated that pickling is a moderate processing technique for reducing mandipropamid residues from vegetable samples. Peeling and juicing were able to effectively remove the mandipropamid residue from the tomato and cucumber samples. For Chinese cabbage and cowpea, r the concentrations of mandipropamid were reduced by > 81% after boiling and steaming for 5 min. After these post-processing treatments, there was no significant enantioselectivity of mandipropamid in the processed vegetable samples. HQ values < 100% demonstrated that the health risks of mandipropamid in four types of processed vegetables could be negligible for Chinese consumers. The results not only provide information about the impact of mandipropamid on food safety, but also help screen contaminated agricultural products for processing methods.”.

Reviewer 2 Report (New Reviewer)

The authors investigated the removal efficiency of different processing methods on mandipropamid residues in contaminated tomato, cucumber, Chinese cabbage and cowpea samples. Washing and soaking reduced the concentrations of mandipropamid in vegetables by half. Pickling was able to reduce mandipropamid residues by a few percent. Juicing and peeling are the most effective processing approaches to remove mandipropamid residues from Chinese cabbage and cowpea samples. The research is interesting and the paper is well organized. Therefore, I suggested that this manuscript may be accepted after a minor revision.

1. To make it easier to understand, please revise the sentence “Mandipropamid, which can effective prevention from most foliar oomycete pathogens, is a chiral systemic fungicide with a broad bactericidal spectrum, and its market sales have been growing since the last half decade [13].”.

2. Line 61 page 2, change “becames” to “became”.

3. Line 108 and line 115 page 3, change “mandipropamid-treated” to “mandipropamid-contaminated”.

4. Line 125-127 page 4, change “The mandipropamid-contaminanted cucumber samples were cut into small pieces, and the mandipropamid-contaminated Chinese cabbage and cowpea samples were drained as the raw material of sauerkraut [22,23].” to “The contaminated cucumber samples were cut into small pieces, and the contaminated Chinese cabbage and cowpea samples were drained as the raw material of sauerkraut [22,23].”.

5. Line 133 page 4, change “mandipropamid-treated” to “mandipropamid-contaminated”.

6. Revise the “Conclusions” section and make it more concise.

7. Check the format of “References” section in accordance with the guidelines of IJERPH.

Author Response

Dear reviewer,

On behalf of all co-authors, I would like to thank you very much for the constructive comments on our manuscript. We have modified our manuscript following your precious recommendations and addressed every comment to the best of our knowledge. The changes are highlighted in red in the revised manuscript. We herein submit the revised manuscript to be considered for publication. Thank you for your helpful coordination and understanding throughout the evaluation process of our manuscript. All Changes Refer to the Previously Submitted Version of the Manuscript.

Reviewers' comments:

Reviewer 2#

The authors investigated the removal efficiency of different processing methods on mandipropamid residues in contaminated tomato, cucumber, Chinese cabbage and cowpea samples. Washing and soaking reduced the concentrations of mandipropamid in vegetables by half. Pickling was able to reduce mandipropamid residues by a few percent. Juicing and peeling are the most effective processing approaches to remove mandipropamid residues from Chinese cabbage and cowpea samples. The research is interesting and the paper is well organized. Therefore, I suggested that this manuscript may be accepted after a minor revision.

  1. To make it easier to understand, please revise the sentence “Mandipropamid, which can effective prevention from most foliar oomycete pathogens, is a chiral systemic fungicide with a broad bactericidal spectrum, and its market sales have been growing since the last half decade [13].”

Answer: Thank you for your suggestion. We have rewritten this sentence. On line 58-60 of page 2 in the previous paper, “Mandipropamid, which can effective prevention from most foliar oomycete pathogens, is a chiral systemic fungicide with a broad bactericidal spectrum, and its market sales have been growing since the last half decade [13].” was changed to “Mandipropamid is a chiral systemic fungicide with a broad bactericidal spectrum and can effective prevention from most foliar oomycete pathogens. The market sales of mandipropamid have been growing since the last half decade [13].”.

  1. Line 61 page 2, change “becames” to “became”.

Answer: Thank you for your suggestion. On line 61 of page 2 in the previous paper, “becames” was changed to “became”.

  1. Line 108 and line 115 page 3, change “mandipropamid-treated” to “mandipropamid-contaminated”.

Answer: Thank you for your suggestion. On line 108 and 115 of page 3 in the previous paper, “mandipropamid-treated” were changed to “mandipropamid-contaminated”.

  1. Line 125-127 page 4, change “The mandipropamid-contaminanted cucumber samples were cut into small pieces, and the mandipropamid-contaminated Chinese cabbage and cowpea samples were drained as the raw material of sauerkraut [22,23].” to “The contaminated cucumber samples were cut into small pieces, and the contaminated Chinese cabbage and cowpea samples were drained as the raw material of sauerkraut [22,23].”

Answer: Thank you for your suggestion. On line 125-127 of page 4 in the previous paper, “The mandipropamid-contaminanted cucumber samples were cut into small pieces, and the mandipropamid-contaminated Chinese cabbage and cowpea samples were drained as the raw material of sauerkraut [22,23].” was changed to “The contaminated cucumber samples were cut into small pieces with an average size of approximately 1 cm3, and the contaminated Chinese cabbage and cowpea samples were drained as the raw material of sauerkraut [20-23].”.

  1. Line 133 page 4, change “mandipropamid-treated” to “mandipropamid-contaminated”.

Answer: Thank you for your suggestion. On line 113 of page 4 in the previous paper, “mandipropamid-treated” was changed to “mandipropamid-contaminated.”

  1. Revise the “Conclusions” section and make it more concise.

Answer: Thank you for your suggestion. The “Conclusion” section was changed to “In this study, the effects of different processing techniques, including washing, soaking, pickling, peeling, juicing, boiling, and steaming, on mandipropamid residues were investigated in four types of vegetables. The results showed that the concentration of mandipropamid decreased after the above processing procedure. The addition of 10% AA, 10% Na2CO3, and 10% NaCl significantly increased the removal efficiency of mandipropamid compared to tap water washing and soaking. The PF values (0.178-0.459) demonstrated that washing and soaking with the acidic solution reduced the mandipropamid residue more than the other three solutions. The decrease in PF from 0.938 to 0.348 illustrated that pickling is a moderate processing technique for reducing mandipropamid residues from vegetable samples. Peeling and juicing were able to effectively remove the mandipropamid residue from the tomato and cucumber samples. For Chinese cabbage and cowpea, r the concentrations of mandipropamid were reduced by > 81% after boiling and steaming for 5 min. After these post-processing treatments, there was no significant enantioselectivity of mandipropamid in the processed vegetable samples. HQ values < 100% demonstrated that the health risks of mandipropamid in four types of processed vegetables could be negligible for Chinese consumers. The results not only provide information about the impact of mandipropamid on food safety, but also help screen contaminated agricultural products for processing methods.”.

  1. Check the format of “References” section in accordance with the guidelines of IJERPH.

Answer: Thank you for your suggestion. We have rechecked the format of “References” section based on the requirement of IJERPH.

Reviewer 3 Report (New Reviewer)

Mandipropamid is a chiral fungicide which has a broad bactericidal spectrum. Due to its good biological activity, it will be widely used in the growing stage of crops. Potential pesticide residue problems could threaten ecosystems and human health. Household processing methods are general techniques used to remove pesticide residue from crops. The authors investigated the removal efficiency of residues of mandipropamid in vegetables by different household processing techniques and found some useful and effective methods to reduce the levels of mandipropamid in vegetables. The paper is interesting and may attract the attention of potential readers. Minor revisions should be made before publication.

1. Please change the improper descriptions, such as becames (line 61 in page 2), pretreatment (line 96 in page 3), sample (line 231 and 232 in page 9), concentration of mandipropamid residue (line 374-375 in page 14).

2. Why did the authors choose the four types of vegetables? Please explain.

3. Why did the authors apply mandipropamid at 140.4 g a.i/hm2(line 100 in page 3)Please explain.

According to ICAMA Pesticide Information Network, mandipropamid has not been registered for use on cucumber,Chinese cabbage, and cowpea.

4. Different household processing techniques have been applied to the removal of mandipropamid from different kinds of vegetable samples. Please state the reason.

4. Delete the sentence “According to the JMPR reports [14], the acute risk of mandipropamind could be negligible for consumers.” (line 292-294 in page 12), which has been mentioned in the “2.4 Health risk estimation” section.

5. Please rewrite the sentence “For Chinese cabbage and cowpea, boiling and steaming for 5 min reduced the concentrations of mandipropamid by > 81%.” (line 387-388 in page 14).

6. Please check the format of the references based on the guidelines.

Author Response

Dear reviewer,

On behalf of all co-authors, I would like to thank you very much for the constructive comments on our manuscript. We have modified our manuscript following your precious recommendations and addressed every comment to the best of our knowledge. The changes are highlighted in red in the revised manuscript. We herein submit the revised manuscript to be considered for publication. Thank you for your helpful coordination and understanding throughout the evaluation process of our manuscript. All Changes Refer to the Previously Submitted Version of the Manuscript.

Reviewers' comments:

Reviewer 3#

Mandipropamid is a chiral fungicide which has a broad bactericidal spectrum. Due to its good biological activity, it will be widely used in the growing stage of crops. Potential pesticide residue problems could threaten ecosystems and human health. Household processing methods are general techniques used to remove pesticide residue from crops. The authors investigated the removal efficiency of residues of mandipropamid in vegetables by different household processing techniques and found some useful and effective methods to reduce the levels of mandipropamid in vegetables. The paper is interesting and may attract the attention of potential readers. Minor revisions should be made before publication.

  1. Please change the improper descriptions, such as becames (line 61 in page 2), pretreatment (line 96 in page 3), sample (line 231 and 232 in page 9), concentration of mandipropamid residue (line 374-375 in page 14).

Answer: Thank you for your suggestion. The changes were listed as follows:

On line 61 of page 2 in the previous paper, “becames” was changed to “became”.

On line 96 of page 3 in the previous paper, “pretreatment” was changed to “pre-treatment”.

On line 231 and 232 of page 9), in the previous paper, “sample” was changed to “samples”.

On line 374-375 of page 14 in the previous paper, “concentration of mandipropamid residue” was changed to “concentration of mandipropamid”.

  1. Why did the authors choose the four types of vegetables? Please explain.

Answer: Thank you for your suggestion. Vegetables, which contain many of the nutrients needed by the human body, are popular, as are tomatoes, cucumbers, cowpeas and the vegetable category. Mandipropamid will be used frequently in these vegetables. Therefore, to better protect these four vegetables, it is important to evaluate these four vegetables through household processing.

  1. Why did the authors apply mandipropamid at 140.4 g a.i/hm2(line 100 in page 3)?Please explain.

According to ICAMA Pesticide Information Network, mandipropamid has not been registered for use on cucumber, Chinese cabbage, and cowpea.

Answer: Thank you for your suggestion. The application doss, spraying time, and interval are obtained from the label of 23.4% mandipropamid SC (Syngenta Nantong Crop Protection Co., Ltd.). For the prevention of late blight, an application dose of 105.3-140.4 g a.i./hm2 was used, with two and three application times, and an interval of 7-10 d. Therefore, we used the highest application dose, most application time, and shortest interval to present the worst conditions. Although mandipropamind has not been registered for use on cucumbers, Chinese cabbage and cowpeas in China, it will be used to control many diseases that may threaten the growth of vegetables including cucumbers, Chinese cabbage and cowpeas due to the better bioactivity of mandipropamid. Therefore, we have investigated the dissipation and distribution of mandipropamid in these vegetables.

  1. Different household processing techniques have been applied to the removal of mandipropamid from different kinds of vegetable samples. Please state the reason.

Answer: Thank you for your suggestion. For certain vegetables, the household processing techniques are chosen based on the common handling in our daily lives. Tomatoes and cucumbers, for example, can be eaten straight after washing or peeling. Chinese cabbages and cowpeas are often consumed after cooking.

  1. Delete the sentence “According to the JMPR reports [14], the acute risk of mandipropamind could be negligible for consumers.” (line 292-294 in page 12), which has been mentioned in the “2.4 Health risk estimation” section.

Answer: Thank you for your suggestion. On lines 292-294 of page 12 in the previous paper, “According to the JMPR reports [14], the acute risk of mandipropamind could be negligible for consumers.” was deleted.

  1. Please rewrite the sentence “For Chinese cabbage and cowpea, boiling and steaming for 5 min reduced the concentrations of mandipropamid by > 81%.” (line 387-388 in page 14).

Answer: Thank you for your suggestion. On line 387-388 of page 14 in the previous paper, “or Chinese cabbage and cowpea, boiling and steaming for 5 min reduced the concentrations of mandipropamid by > 81%.” was changed to “For Chinese cabbage and cowpea, r the concentrations of mandipropamid were reduced by > 81% after boiling and steaming for 5 min.”

  1. Please check the format of the references based on the guidelines.

Answer: Thank you for your suggestion. We have rechecked the format of “References” section based on the requirement of IJERPH.

Round 2

Reviewer 1 Report (New Reviewer)

The document has been corrected according to my previous comments. I have no further suggestions, and I think the manuscript could be published in its present form.

This manuscript is a resubmission of an earlier submission. The following is a list of the peer review reports and author responses from that submission.

Round 1

Reviewer 1 Report

Dear authors

To provide more support for your data your statistical analysis must be more robust. The n value is only 3. Besides, you must give a better description of the methodology and discussion section. Figure 1 should be part of the MS, not as supplementary material. Most of the study is referred to supplementary data.

The English must be reviewed by a native speaker. You can find more comments in the attached file. 

Reviewer 2 Report

This work evaluated the effects of household processing on residues of the chiral fungicide Mandipropamid in four kinds of vegetable samples.

This work presents in a clear and objective way all its development as material, methods, results and conclusions.

However, it is a very specific demand about Chinese consumers and does not present any new information or innovation.

This can be observed since all the techniques and methodologies used, such as liquid chromatography with tandem mass spectrometry, data calculation,  health risk estimation, hazard quotient (HQ) values and statistical analysis are common and widely known.